# Cross-contamination effect on turbulence spectra from Doppler beam swinging wind lidar

Felix Kelberlau[1] and Jakob Mann[2]

[1]NTNU, Department of Energy and Process Engineering, Norwegian University of Science and Technology, 7491 Trondheim, Norway
[2]DTU Wind Energy, Technical University of Denmark, 4000 Roskilde, Denmark

**Correspondence:** Felix Kelberlau (felix.kelberlau@ntnu.no) and Jakob Mann (jmsq@dtu.dk)

**Abstract.** Turbulence velocity spectra are of high importance for the estimation of loads on wind turbines and other built structures, as well as for fitting measured turbulence values to turbulence models. Spectra generated from reconstructed wind vectors of Doppler beam swinging (DBS) wind lidars differ from spectra based on one-point measurements. Profiling wind lidars have several characteristics that cause these deviations, namely cross-contamination between the three velocity components, averaging along the lines-of-sight, and the limited sampling frequency. This study focuses on analyzing the cross-contamination effect. We sample wind data in a computer generated turbulence box to predict lidar derived turbulence spectra for three wind directions and four measurement heights. The data are then processed with the conventional method and with the method of squeezing that reduces the longitudinal separation distances between the measurement locations of the different lidar beams by introducing a time lag into the data processing. The results are analyzed and compared to turbulence velocity spectra from field measurements with a Windcube V2 wind lidar and ultrasonic anemometers as reference. We successfully predict lidar derived spectra for all test cases and found that their shape is dependent on the angle between the wind direction and the lidar beams. With conventional processing, cross-contamination affects all spectra of the horizontal wind velocity components. The method of squeezing improves the spectra to an acceptable level only for the case of the longitudinal wind velocity component and when the wind blows parallel to one of the lines-of-sight. The analysis of the simulated spectra described here improves our understanding of the limitations of turbulence measurements with DBS profiling wind lidar.

## 1 Introduction

Wind energy research and industry depend on reliable measurements of wind velocities for wind site assessment and load prediction. Remote sensing devices such as vertical profiling lidars can measure wind velocities at adjustable height levels from the ground. The ease of installation and mobility of ground-based lidars make them superior to conventional in-situ anemometry on tall meteorological masts.

Vertical profiling wind lidars emit a laser beam into different directions and can estimate the radial component of the wind velocity along sections of the beam. Measurements of the radial velocity in at least three different directions are then used to reconstruct three-dimensional wind vectors. Depending on the type of lidar being applied either velocity-azimuth display (VAD) scanning or Doppler beam swinging (DBS) is used as the scanning strategy. When VAD scanning is applied, the laser beam performs continuous azimuth scans at a fixed elevation angle (Browning and Wexler, 1968). With DBS the beam is directed into certain directions where it accumulates measurement data for a defined time before it swings into the next direction. Turbulence statistics can be derived from VAD scanning (e.g. Eberhard et al., 1989; Krishnamurthy et al., 2011; Smalikho, 2003) or DBS (e.g. Frehlich et al., 1998; Kumer et al., 2016; Bodini et al., 2019). An advantage of DBS is that the signal-to-noise ratio of each radial velocity estimate increases with accumulation time in each direction. The possibility to measure into a vertical direction is another advantage of DBS wind lidars. The Windcube produced by Leosphere (Saclay, France) is a widely used vertical profiling pulsed Doppler wind lidar that uses DBS to reconstruct three-dimensional wind vectors from five independent line-of-sight (LOS) velocity measurements.

Profiling lidars have proven to be accurate tools for measuring mean wind speed and direction in non-complex terrain (Emeis et al., 2007; Smith et al., 2006; Gottschall et al., 2012; Kim et al., 2016). But the measurement of turbulence with ground based profiling wind lidars is inaccurate due to their extended measurement volumes, the limited sampling frequency for each line-of-sight measurement, and the large spatial separation between the measurement volumes (Sathe and Mann, 2013; Newman et al., 2016). The second-order statistics of turbulence measured by profiling wind lidar show that the measurement error depends on several factors: the measurement principle of the lidar used, the conditions of the atmospheric boundary layer, the measurement height, and in the case of the Windcube also on the angle between the mean wind direction and the orientation of the lidar beams (Sathe et al., 2011).

Measured auto- and co-spectra of the three turbulent wind velocity components show the spectral distribution of the wind velocity variance. IEC standard 61400-1 (IEC, 2019) recommends to use such one-point spectra for finding the model parameters anisotropy $\gamma$, length scale $L$, and dissipation factor $\alpha\epsilon^{\frac{2}{3}}$ of the uniform shear model of turbulence (Mann, 1994). This can be done by fitting the parameters to the measured spectra. The found parameters can then be used in the process of determining aerodynamic loads on wind turbines and other built structures. But estimations of turbulence spectra from wind lidar data deviate significantly from reference measurements taken at meteorological masts due to their measurement principle. Canadillas et al. (2010) present measured turbulence velocity spectra from a Windcube that show characteristic differences in comparison to reference measurements from sonic anemometers. The lidar spectra show e.g. too high spectral energies in a wide range of frequencies range due to cross-contamination and gaps at frequencies that correspond to the limited sampling frequency of the lidar beams. Such spectra are modeled in Sathe and Mann (2012) for an older Windcube version. The same model can, with minor modifications, be used to predict spectra from the current version of the Windcube that samples faster and includes a vertical beam. The major drawback of the model is that it cannot predict spectra for cases in which the wind inflow is not parallel to two of the lidar beams.

In the study we present here, we overcome this limitation by sampling velocity values in a computer-generated turbulence box and processing them in a similar fashion to how DBS scanning pulsed lidar samples wind velocities in the atmosphere.

The results of this artificial sampling are compared to measured DBS pulsed lidar spectra acquired from field measurements. This method makes it possible to predict lidar derived turbulence velocity spectra for all relative wind directions.

In addition to conventional DBS processing of radial wind velocities, we reconstruct the three-dimensional wind vectors with the method of squeezing introduced in Kelberlau and Mann (2019). This method minimizes cross-contamination for VAD scanning wind lidars (e.g., ZX 300) by introducing a time lag into the data processing that compensates for the duration it takes to advect an air volume from one lidar beam to the other.

In this study, we assess whether the method of squeezing is advantageous also for DBS scanning wind lidar such as the Windcube and to what extent it improves estimation of turbulence velocity spectra. The aim of the work presented here is prediction of turbulence velocity spectra from DBS scanning wind lidars and making turbulence measurements more accurate by applying a modified data processing algorithm.

Next, section 2 presents the theory of how a pulsed Doppler beam swinging wind lidar determines radial wind velocities and reconstructs three-dimensional wind vectors. The method of squeezing is also briefly presented. In section 3, we describe the methods applied in this study. These consist of, first, field measurements with a Windcube V2 and collocated reference measurements with sonic anemometers on a large meteorological mast and, second, sampling of computer generated turbulence data. We present and discuss the results of both field measurements and simulations in section 4 and describe our key findings in the conclusions section 5. A nomenclature can be found in the appendix.

## 2 Lidar theory

### 2.1 Coordinate system and preliminaries

This study uses a right-handed coordinate system aligned with the horizontal mean wind vector. The component $u$ is pointing into the mean wind direction, $v$ is the transversal wind component, and $w$ points vertically upwards, such that for the wind vector $\boldsymbol{u}$ it accounts

$$\boldsymbol{u} = \begin{bmatrix} u \\ v \\ w \end{bmatrix}. \tag{1}$$

We also use Reynolds decomposition with a time scale of ten minutes to divide the wind vectors into a mean part $\boldsymbol{U}$ and a fluctuating part $\boldsymbol{u}'$, such that

$$\boldsymbol{u} = \boldsymbol{U} + \boldsymbol{u}'. \tag{2}$$

$U$ is the mean wind speed, the transversal component $V$ is by definition zero, and the vertical mean velocity $W$ in non-complex terrain is typically also close to zero. The mean values of the components of $\boldsymbol{u}'$ are by definition zero, but their statistical variance provides important information about the amount of turbulence in the wind. It is defined as

$$\sigma_u^2 = \langle u'u' \rangle, \tag{3}$$

where $\langle\rangle$ means ensemble averaging. The variance of the other two components $\sigma_v^2$ and $\sigma_w^2$ can be calculated accordingly.

## 2.2 Line-of-sight velocity retrieval

The Windcube lidar emits laser beams into five fixed directions. As shown in Fig. 1 four beams are inclined by the zenith angle $\phi$ from the vertical and separated along the horizon by the azimuth angle $\theta$. The fifth beam points vertically upwards. The beam directions define the internal fixed right-handed coordinate system of the Windcube. In accordance with the documentation of the Windcube, the $x$-component is oriented from LOS1 towards LOS3, the $y$-component points from LOS2 towards LOS4, and the vertical $z$-component points downwards along LOS5. In the default setup, the LOS1 beam is oriented towards north. If this is not the case, a directional offset $\theta_0$ must be considered in the data processing. Unit vectors $\boldsymbol{n}$ that point into the direction of the five beams are defined as

$$\boldsymbol{n}_i = \begin{bmatrix} \cos\left(\frac{i-3}{2}\pi\right)\sin\phi \\ \sin\left(\frac{i-3}{2}\pi\right)\sin\phi \\ -\cos\phi \end{bmatrix} \text{ for } i = 1...4 \text{ and } \boldsymbol{n}_5 = \begin{bmatrix} 0 \\ 0 \\ -1 \end{bmatrix} \tag{4}$$

A small portion of the emitted laser radiation is backscattered in the direction of origin. This backscattered radiation has a wavelength that is slightly different from the emitted radiation. The difference in wavelength is caused by the Doppler effect and is proportional to the component of the wind in the respective beam direction which is

$$v_{r_i} = \boldsymbol{n}_i \cdot \boldsymbol{x}_i \tag{5}$$

where $\boldsymbol{x}_i$ is the wind velocity vector at the measurement points in the coordinate system of the Windcube. The Doppler shift can be detected and is used to determine the line-of-sight velocities, i.e., the radial velocities in the corresponding beam direction. Unlike continuous-wave lidars, pulsed lidars can determine signed line-of-sight velocities for multiple height levels simultaneously. These line-of-sight velocities are the weighted average of the radial wind velocities along the stretch of the lidar beam that is illuminated by the range gate. A reasonable weighting function to model the line-of-sight averaging is the convolution of the laser pulse shape with the interrogation window. In the case of the Windcube, the emitted laser pulses are $175\,\mathrm{ns}$ long and thus illuminate air volumes of $175\,\mathrm{ns} \times c = 52.46\,\mathrm{m}$ in length along the line-of-sight, where $c$ is the speed on light. The backscattered radiation recorded by the laser detector at one point in time originates from a line-of-sight segment that cannot be shorter than half of this length. If the laser beam were perfectly collimated and rectangular and an interrogation window of the same length were chosen, a triangular function would be the correct weighting function to account for the higher likeliness of a scatterer to be located closer to the center of the pulse than its ends. However, the beams of the Windcube not collimated but focused permanently to a height level of approximately $100\,\mathrm{m}$ in order to optimize the carrier-to-noise ratio. In addition, its light pulses are not perfectly cut-in and -out at their ends. The triangular function is thus only an approximation of the real situation. We refer to Lindelöw (2008) for more details. However, as in Sathe and Mann (2012), we use a triangular weighting function

$$\varphi(s) = \frac{l_p - |s|}{l_p^2} \text{ for } |s| < l_p \text{ and } \varphi(s) = 0 \text{ for } |s| \geq l_p \tag{6}$$

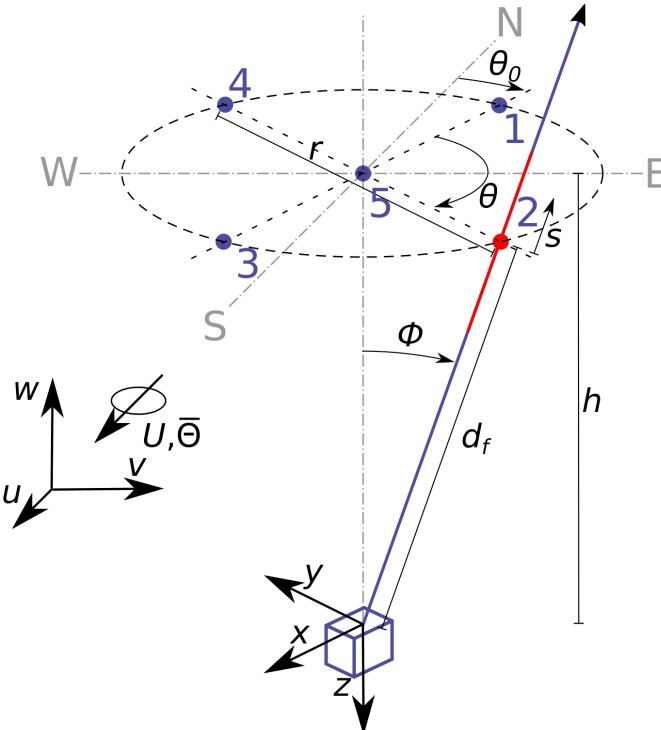

**Figure 1.** Visualization of the beam configuration of the Windcube V2, relevant lengths and angles, as well as the two coordinate systems used by the lidar and in wind data analysis. For better visibility, only LOS2 is depicted as a beam with the range gate indicated in red along the blue laser beam.

where $s$ is the distance from the midpoint of the range gate and $l_p = 26\,\mathrm{m}$ is the approximate half length of the range gate to simulate the lidar derived weighted radial velocity

$$\tilde{v}_{r_i} = \int\limits_{-\infty}^{\infty} \varphi(s)\boldsymbol{n_i} \cdot \boldsymbol{u}((s+d_f)\boldsymbol{n_i})ds \tag{7}$$

where $d_f$ is the distance of the center of the range gate from the lidar.

## 2.3 DBS measurement principle

The line-of-sight velocities are processed in order to reconstruct three-dimensional wind vectors. These are based on the fixed right-handed coordinate system of the Windcube. The Windcube calculates one new wind vector component whenever a new line-of-sight measurement becomes available. The $x$-component is calculated when a radial velocity of either LOS1 or LOS3 is retrieved. The newly updated line-of-sight velocity is then combined with the immediate precursor of the opposing direction according to

$$x = \frac{\tilde{v}_{r_1} - \tilde{v}_{r_3}}{2\sin\phi}. \tag{8}$$

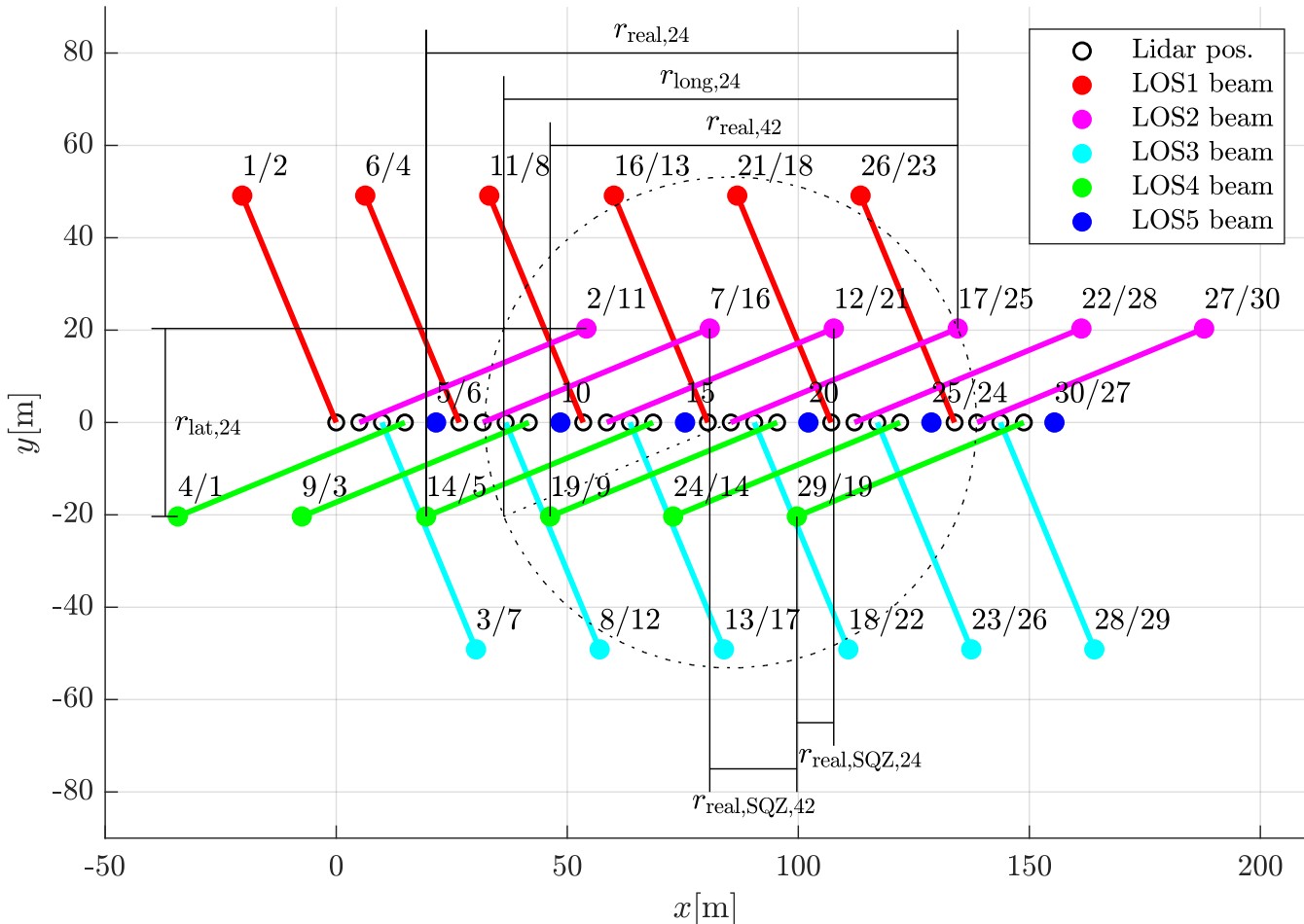

**Figure 2.** Visualization of the measurement geometry of the Windcube V2 with the five beam directions, LOS1-LOS5 (color coded). Top view of 30 consecutive line-of-sight measurements in a coordinate system that is moving with the mean wind. The angle between the mean wind and the LOS1–LOS3 axis is $\alpha = 67.5°$. Measurement locations (dots) are numbered by their order in time (first number) and position in wind direction (second number). Longitudinal and lateral separation distances for combinations of LOS2 and LOS4 beams are shown.

The $y$-component is calculated from LOS2 and LOS4 according to

$$y = \frac{\tilde{v}_{r_2} - \tilde{v}_{r_4}}{2\sin\phi}. \tag{9}$$

Here, the latest LOS2 beam is combined with the previous LOS4 beam and vice versa. In Fig. 2 it can be seen that e.g. the measurement of the 17th beam that the lidar emits (LOS2) is combined with the 14th beam (LOS4) and the 19th beam (LOS4) is combined with the 17th beam (LOS2) to calculate two values of $y$.

| LOS # | $\phi$ | $\theta$ | $t$ | $\Delta t$ |
|---|---|---|---|---|
| 1 | 28° | 0° | 0.00s | — |
| 2 | 28° | 90° | 0.72s | 0.72s |
| 3 | 28° | 180° | 1.44s | 0.72s |
| 4 | 28° | 270° | 2.16s | 0.72s |
| 5 | 0° | - | 3.13s | 0.97s |
| 1 | 28° | 0° | 3.85s | 0.72s |
| ⋮ | ⋮ | ⋮ | ⋮ | ⋮ |

**Table 1.** Line-of-sight beam geometry and timing. $t$ is the accumulated time after the first beam measurement and $\Delta t$ is the time difference between the current and the previous beam measurement.

The vertical $z$-component can be estimated directly from the vertical beam result whenever a new LOS5 measurement becomes available so that

$$z = \tilde{v}_{r_5}. \tag{10}$$

In addition to the three wind components, the Windcube estimates the horizontal wind velocity

$$V_{\mathrm{hor}} = \sqrt{x^2 + y^2}, \tag{11}$$

the horizontal wind direction clockwise from north

$$\Theta = \theta_0 - \arctan(y, -x) \tag{12}$$

and their ten-minute average values $\overline{V}_{\mathrm{hor}}$ and $\overline{\Theta}$ marked with an overline.

In order to rotate the three wind vector components into the coordinate system aligned with the mean wind direction, we calculate

$$\boldsymbol{u}_{\mathrm{DBS}} = \begin{bmatrix} u_{\mathrm{DBS}} \\ v_{\mathrm{DBS}} \\ w_{\mathrm{DBS}} \end{bmatrix} = \begin{bmatrix} x\cos\alpha + y\sin\alpha \\ x\sin\alpha - y\cos\alpha \\ -z \end{bmatrix} \tag{13}$$

where $\alpha = \overline{\Theta} - \theta_0$ is the relative inflow angle. The resulting wind vectors are updated at slightly varying times because swinging the Doppler beam from one line-of-sight to the next and accumulate measurements takes approximately $0.72\,\mathrm{s}$ for the inclined beams and $0.97\,\mathrm{s}$ for the vertical beam. We do not know the reason for the different times required to change the beam direction. This leads to an average wind vector refresh rate of approximately $1.3\,\mathrm{Hz}$ although each beam is updated with a frequency of no more than $0.26\,\mathrm{Hz}$. Table 1 provides an overview of the beam geometry and the timing.

## 2.4 Measurement errors due to cross-contamination

The $w$-component is measured directly from the vertical beam. However, the reconstruction of the horizontal wind components $u$ and $v$ involves the combination of measurement values from two spatially separated air volumes. These reconstructions are correct only if the wind vector is identical at all measurement volumes. For the calculation of average wind speeds, it is sufficient that the average wind vector is identical at all measurement volumes. But for every single wind vector to be correct, the wind field would need to be static. In a turbulent wind field, the single reconstructed wind vectors are erroneous due to cross-contamination of the different wind velocity components.

The cause of this error lies in combining radial velocities from spatially separated air volumes. The separations can be categorized into longitudinal separations (along the direction of the mean wind) and lateral separations (orthogonal to the mean wind direction). Assuming Taylor's frozen turbulence hypothesis (Taylor, 1938), wind velocities sampled at two longitudinally separated points are perfectly correlated but have a temporal offset between the two measurement signals that corresponds to the time needed for the mean wind speed to cover the distance between the two points. Whenever the wavelength of the measured turbulence equals $2/n$ times the separation distance, with $n = 1, 3, 5...$, a resonance effect occurs. The wind speed component being measured cannot be detected in these cases and is replaced by contributions of other wind speed components. In contrast, for $n = 0, 2, 4...$ no resonance effect occurs (see Fig. 2 in Kelberlau and Mann (2019)).

The distance $D$ between two opposing measurement points is

$$D = 2h \tan \phi \tag{14}$$

where $h$ is the measurement height. $D$ is the diameter of the dotted circle in Fig. 2. The longitudinal separation distances for the beam combination LOS1 and LOS3 can be calculated according to

$$r_{\mathrm{long},13} = |D \cos \alpha| \tag{15}$$

$r_{\mathrm{long},24}$ for the beam combination LOS2 and LOS4 can be estimated by swapping the cosine in eq. 15 by a sine. $r_{\mathrm{long},24}$ is also shown in Fig. 2.

Eq. 13 shows that the components $u$ and $v$ in the reconstructed wind vectors are composed of contributions from two different beam combinations. These are LOS1 and LOS3 (see Eq. 8) as well as LOS2 and LOS4 (see Eq.9). In order to calculate longitudinal separations that are representative for the reconstructed wind velocity components we must introduce a weighting and calculate

$$r_{\mathrm{rep},u} = \frac{|\cos \alpha| \times r_{\mathrm{long},13} + |\sin \alpha| \times r_{\mathrm{long},24}}{|\cos \alpha| + |\sin \alpha|} = \frac{D}{|\cos \alpha| + |\sin \alpha|} \tag{16}$$

for the $u$ component and

$$r_{\mathrm{rep},v} = \frac{|\sin \alpha| \times r_{\mathrm{long},13} + |-\cos \alpha| \times r_{\mathrm{long},24}}{|\cos \alpha| + |\sin \alpha|} = \frac{|\sin (2\alpha)| D}{|\cos \alpha| + |\sin \alpha|} \tag{17}$$

for the $v$ component. The resulting representative longitudinal separation distance values for the Windcube for four measurement heights $40\,\mathrm{m}$, $60\,\mathrm{m}$, $80\,\mathrm{m}$, and $100\,\mathrm{m}$ and for three relative wind inflow angles $\alpha = 0°$, $22.5°$, and $45°$ are given in table 2.

| $h$ | $\alpha = 0°$ | | $\alpha = 22.5°$ | | $\alpha = 45°$ | |
|---|---|---|---|---|---|---|
| | $r_{\mathrm{rep},u}$ | $r_{\mathrm{rep},v}$ | $r_{\mathrm{rep},u}$ | $r_{\mathrm{rep},v}$ | $r_{\mathrm{rep},u}$ | $r_{\mathrm{rep},v}$ |
| 40 | 42.5 | 0.0 | 32.6 | 23.0 | 30.1 | 30.1 |
| 60 | 63.8 | 0.0 | 48.8 | 34.5 | 45.1 | 45.1 |
| 80 | 85.1 | 0.0 | 65.1 | 46.0 | 60.2 | 60.2 |
| 100 | 106.3 | 0.0 | 81.4 | 57.6 | 75.2 | 75.2 |

**Table 2.** Representative longitudinal separation distances influencing the $u$ and $v$-component of $\boldsymbol{u}_{\mathrm{DBS}}$ for all investigated test cases. All values given in [m].

From these distances, the wave numbers at which we expect resonance can easily be determined with $k_{\mathrm{res}} = n\pi/r_{\mathrm{rep}}$ where $n$ is an odd integer. Lateral separation distances $r_{\mathrm{lat},ij}$ could be estimated in a similar way. But compared to longitudinal separations, the situation is different for wind velocity fluctuations measured at two laterally separated points. The spatial structure of turbulence leads to the wind velocity fluctuations becoming less correlated as the distance between the two measurement points increases. The coherence of the fluctuations is also weaker for small eddies than for large turbulent structures. That means that a turbulent structure can only be detected at two laterally separated points if the length scale of the turbulent structure is large compared to the separation distance. Lateral separation leads to contamination that occurs gradually without resonance points at specific wave numbers.

If the mean wind is aligned with two opposing lines-of-sight, e.g., blows in the LOS1 – LOS3 direction, then the $u$-component of the wind vector is reconstructed from two points that are only separated longitudinally. That means each turbulent structure is measured twice: once, when it passes the LOS1 location, and then some time later at the LOS3 location. Assuming frozen turbulence, measurements from points that are separated longitudinally are fully correlated, and resonance occurs at specific wave numbers. The $v$-component, on the contrary, is in this case reconstructed from the laterally separated points of LOS2 and LOS4, and a reduced correlation is found depending on the size of the turbulent structure and the separation distance. No specific resonance wave numbers are found. For a comprehensive description of the cross-contamination effects due to isolated longitudinal and isolated lateral separation, see Kelberlau and Mann (2019). Here we look at the more complex case when the mean wind inflow is not aligned with two opposing line-of-sight directions. Estimates of one horizontal wind velocity component can then be contaminated by contributions from both other wind velocity components. For a manual estimation of the cross-contamination effect for non-aligned inflow we first derive the lidar estimated wind vector component $u_{\mathrm{DBS}}$ as a function of the real wind vector at all four measurement locations. When, Eqs. 8 and 9 are set into Eq. 13 we get

$$u_{\mathrm{DBS}} = \frac{(\tilde{v}_{r_1} - \tilde{v}_{r_3})\cos\alpha}{2\sin\phi} + \frac{(\tilde{v}_{r_2} - \tilde{v}_{r_4})\sin\alpha}{2\sin\phi}. \tag{18}$$

We assume no line-of-sight averaging, so that $v_{r_i} = \tilde{v}_{r_i}$ and use Eqs. 4 and 5. After rearranging we get

$$u_{\mathrm{DBS}} = \frac{\cos\alpha}{2}(-x_1 + z_1\cot\phi - x_3 - z_3\cot\phi) + \frac{\sin\alpha}{2}(-y_2 + z_2\cot\phi - y_4 - z_4\cot\phi). \tag{19}$$

After transferring the wind velocity components $x,y,z$ into the $u,v,w$ coordinate system we get

$$u_{\text{DBS}} = \frac{\cos\alpha}{2}(-u_1\cos\alpha - v_1\sin\alpha - w_1\cot\phi - u_3\cos\alpha - v_3\sin\alpha + w_3\cot\phi)$$
$$+ \frac{\sin\alpha}{2}(-u_2\sin\alpha + v_2\cos\alpha - w_2\cot\phi - u_4\sin\alpha + v_4\cos\alpha + w_4\cot\phi). \tag{20}$$

With Eq. 3 we can describe the total lidar variance as a function of the wind vector fluctuations at the four measurement points as

$$\sigma_{u,\text{DBS}}^2 = \langle u_{\text{DBS}}'^2 \rangle = \frac{1}{4}\Big\langle \Big( \big(u_1'\cos\alpha + v_1'\sin\alpha + w_1'\cot\phi + u_3'\cos\alpha + v_3'\sin\alpha - w_3'\cot\phi\big)\cos\alpha$$
$$+ \big(u_2'\sin\alpha - v_2'\cos\alpha + w_2'\cot\phi + u_4'\sin\alpha - v_4'\cos\alpha - w_4'\cot\phi\big)\sin\alpha \Big)^2 \Big\rangle. \tag{21}$$

A similar formula can be found for the transversal component

$$\sigma_{v,\text{DBS}}^2 = \langle v_{\text{DBS}}'^2 \rangle = \frac{1}{4}\Big\langle \Big( \big(u_1'\cos\alpha + v_1'\sin\alpha + w_1'\cot\phi + u_3'\cos\alpha + v_3'\sin\alpha - w_3'\cot\phi\big)\sin\alpha$$
$$- \big(u_2'\sin\alpha - v_2'\cos\alpha + w_2'\cot\phi + u_4'\sin\alpha - v_4'\cos\alpha - w_4'\cot\phi\big)\cos\alpha \Big)^2 \Big\rangle. \tag{22}$$

Power spectral densities $F_{\text{DBS}}$ at particular wave numbers are composed of the same linear combinations of wind components as the total variances in Eqs. 21 and 22. These equations are thus helpful when analyzing the extent of cross contamination at particular wave numbers. As an example, we now take the case when the mean wind direction and one of the lines-of-sight create an angle of 45°. We assume $\overline{\Theta} = 90°$ and $\theta_0 = 45°$ because this situation is found in the measurements described later in this study. However, the results are identical for all setups in which the relative wind inflow $\alpha = 45°$. In this case, LOS4 and LOS3 are separated purely longitudinally from LOS1 and LOS2, and LOS2 and LOS3 are separated purely laterally from LOS1 and LOS4 as shown in Figure 3. This opens up the possibility of determining the cross-contamination effect for four extreme conditions. These four extreme conditions are characterized by either full or no longitudinal resonance as well as either perfect or no lateral correlation. In the first case a) when no resonance occurs and the lateral correlation is perfect, we assume identical wind vectors at all four points. It accounts: $\boldsymbol{u}'_{1,a} = \boldsymbol{u}'_{2,a} = \boldsymbol{u}'_{3,a} = \boldsymbol{u}'_{4,a} = \boldsymbol{u}'_{\text{I}}$. In the second case b) when no resonance occurs but the lateral correlation is zero, it accounts: $\boldsymbol{u}'_{1,b} = \boldsymbol{u}'_{4,b} = \boldsymbol{u}'_{\text{I}}$ and $\boldsymbol{u}'_{2,b} = \boldsymbol{u}'_{4,b} = \boldsymbol{u}'_{\text{II}}$ where $\boldsymbol{u}'_{\text{I}}$ and $\boldsymbol{u}'_{\text{II}}$ are independent vectors. In the third case c) resonance between the longitudinally separated points occurs and the fluctuations at laterally separated points are perfectly correlated. It accounts: $\boldsymbol{u}'_{1,c} = \boldsymbol{u}'_{2,c} = -\boldsymbol{u}'_{3,c} = -\boldsymbol{u}'_{4,c} = \boldsymbol{u}'_{\text{I}}$. The fourth case d) is characterized by longitudinal resonance and zero lateral correlation. It accounts: $\boldsymbol{u}'_{1,d} = -\boldsymbol{u}'_{4,d} = \boldsymbol{u}'_{\text{I}}$ and $\boldsymbol{u}'_{2,d} = -\boldsymbol{u}'_{3,d} = \boldsymbol{u}'_{\text{II}}$ where $\boldsymbol{u}'_{\text{I}}$ and $\boldsymbol{u}'_{\text{II}}$ are independent vectors. Figure 3 gives an overview of the conditions we assume for these four cases a) to d). With these assumptions, Eq. 21 provides the lidar estimates of the power spectral density values $F_{u,\text{DBS}}$ as linear combinations of the spectral values of the three wind components $F_u$, $F_v$ and $F_w$, as shown in the lower half of table 3. The resulting linear combinations of power spectral densities that compose the lidar-measured $u$ and $v$-components of turbulence for the case with $\alpha = 0°$ are shown in the upper half of the same table.

Table 3 can be read as follows. First, choose the aligned ($\alpha = 0°$) or non-aligned case ($\alpha = 45°$). Then select the wind component of interest: $F_{u,\text{DBS}}$ or $F_{v,\text{DBS}}$. Next, decide if the situation with or without resonance is more relevant for the

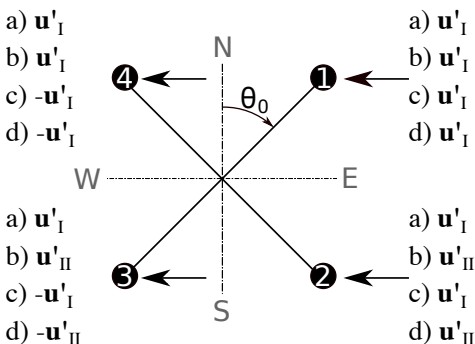

a) No resonance, laterally correlated
b) No resonance, laterally uncorrelated
c) Resonance, laterally correlated
d) Resonance, laterally uncorrelated

a) $\mathbf{u'}_{\mathrm{I}}$          a) $\mathbf{u'}_{\mathrm{I}}$
b) $\mathbf{u'}_{\mathrm{I}}$          b) $\mathbf{u'}_{\mathrm{I}}$
c) $-\mathbf{u'}_{\mathrm{I}}$          c) $\mathbf{u'}_{\mathrm{I}}$
d) $-\mathbf{u'}_{\mathrm{I}}$          d) $\mathbf{u'}_{\mathrm{I}}$

a) $\mathbf{u'}_{\mathrm{I}}$          a) $\mathbf{u'}_{\mathrm{I}}$
b) $\mathbf{u'}_{\mathrm{II}}$          b) $\mathbf{u'}_{\mathrm{II}}$
c) $-\mathbf{u'}_{\mathrm{I}}$          c) $\mathbf{u'}_{\mathrm{I}}$
d) $-\mathbf{u'}_{\mathrm{II}}$          d) $\mathbf{u'}_{\mathrm{II}}$

**Figure 3.** Overview of the assumptions made to determine the cross-contamination values listed in Table 3. In cases with no resonance, the wind vectors $\boldsymbol{u'}_{\mathrm{I,II}}$ are identical at the longitudinally separated measurement points. In resonance cases they have an opposite sign. In cases with laterally correlated velocities, the wind vectors at laterally separated measurement points are identical. And in cases with no correlation at points that are laterally separated the wind vectors $\boldsymbol{u'}_{\mathrm{I}}$ and $\boldsymbol{u'}_{\mathrm{II}}$ are independent.

wave number of interest. Finally, select a block of values that either represents the case with perfect lateral correlation or that assumes laterally uncorrelated fluctuations. The sum of the variances of the wind components multiplied by the values given in this block is the theoretical lidar derived variance of the selected component. It is usually unclear to which degree the fluctuations are correlated but the table can still be used for rough estimations. If you look for example at the resonance
case for $u$ you will find that the lidar does not detect longitudinal wind fluctuations at all, while the lidar estimated $u$-variance $F_{u,\mathrm{DBS}}$ is composed of a weakened $v$-signal of between 0.00 and 0.50 times the real $v$-fluctuations and an amplified $w$-signal of between 3.54 and 7.07 times the real $w$-fluctuations depending on the degree of lateral correlation. The values given in the table can explain many of the effects we later see in the lidar derived spectra for non-aligned inflow.

Table 1 shows that the radial velocity for each line-of-sight is determined not continuously but once every $3.85\,\mathrm{s}$. That means
turbulent fluctuations which occur with a corresponding frequency cannot be detected by any of the Windcube's lidar beams. The respective wave numbers are

$$k_{\mathrm{scan}} = \frac{2\pi}{U \cdot 3.85\,\mathrm{s}}. \tag{23}$$

At these wave numbers $k_{\mathrm{scan}}$ we expect sudden drops in all lidar derived spectra.

Because the data are not acquired continuously we expect a second effect that influences the shape of the lidar derived
turbulence velocity spectra. In the previous subsection we estimated the longitudinal separations (Table 2). These separations represent statistical averages, not actual separations. The actual separations could only be identical to these values if the lidar acquired line-of-sight velocity values continuously, which is not the case. Take the example of wind blowing along the $x$-

$$\alpha = 0°$$

| | $F_{u,\text{DBS}}$ | $F_{v,\text{DBS}}$ | |
| --- | --- | --- | --- |
| | — | lat. corr. | lat. uncorr. |
| | $1.00\,F_u$ | $0.00\,F_u$ | $0.00\,F_u$ |
| no resonance | $0.00\,F_v$ | $1.00\,F_v$ | $0.50\,F_v$ |
| | $0.00\,F_w$ | $0.00\,F_w$ | $1.77\,F_w$ |
| | $0.00\,F_u$ | | |
| resonance | $0.00\,F_v$ | — | — |
| | $3.54\,F_w$ | | |

$$\alpha = 45°$$

| | $F_{u,\text{DBS}}$ | | $F_{v,\text{DBS}}$ | |
| --- | --- | --- | --- | --- |
| | lat. corr. | lat. uncorr. | lat. corr. | lat. uncorr. |
| | $1.00\,F_u$ | $0.50\,F_u$ | $0.00\,F_u$ | $0.00\,F_u$ |
| no resonance | $0.00\,F_v$ | $0.00\,F_v$ | $1.00\,F_v$ | $0.50\,F_v$ |
| | $0.00\,F_w$ | $0.00\,F_w$ | $0.00\,F_w$ | $3.54\,F_w$ |
| | $0.00\,F_u$ | $0.00\,F_u$ | $0.00\,F_u$ | $0.50\,F_u$ |
| resonance | $0.00\,F_v$ | $0.50\,F_v$ | $0.00\,F_v$ | $0.00\,F_v$ |
| | $7.07\,F_w$ | $3.54\,F_w$ | $0.00\,F_w$ | $0.00\,F_w$ |

**Table 3.** Expected contribution of the power spectral densities $F_u$, $F_v$, and $F_w$ of the wind velocity components on the lidar derived values of $F_{u,\text{DBS}}$ and $F_{v,\text{DBS}}$ for aligned and non-aligned inflow with $\alpha = 0°$ and $45°$.

axis from LOS1 to LOS3. When an air volume is measured at LOS1, it continues moving towards LOS3. When the lidar subsequently takes a sample at LOS3, the actual separation distance between these two air volumes is less than the physical distance between the lines-of-sight. Conversely, when an air volume is measured at LOS3 first, it will have advected further away by the time the next sample is taken at LOS1. In this case, the actual separation distance will be larger than the physical distance between LOS1 and LOS3. As in table 1, the time difference of $\Delta t_{13} = 1.44\,\text{s}$ between a measurement of LOS1 and LOS3 deviates from the time difference $\Delta t_{31} = 2.41\,\text{s}$ between measurements at LOS3 and LOS1. The actual separation distances are then

$$r_{\text{real},13} = r_{\text{long},13} + \Delta t_{13} U \tag{24}$$

and $r_{\text{real},31} = r_{\text{long},13} - \Delta t_{31} U$.

The turbulence velocity spectra that we later derive from the lidar measurements can be seen as the average of two types of spectra: the ones we get from reconstructing the wind vector components of only LOS1 with the previous LOS3 measurements and the ones we get from reconstructing the wind vector components of only LOS3 with the previous LOS1 measurement. These averaged spectra deviate significantly from the spectra expected from continuous sampling, if the product of mean wind

speed and the time between the measurements is large compared to the average separation distances. The resonance peaks are then less pronounced and extend over a wider range of wave numbers.

## 2.5 Squeezed wind vector reconstruction

One method to avoid cross-contamination caused by longitudinal separation is presented in Kelberlau and Mann (2019). It is called the method of squeezing and aims at removing the longitudinal separation distances $r_{\mathrm{real},ij}$ by introducing a temporal delay $\tau = \frac{r_{\mathrm{real},ij}}{U}$ into the data processing. The length of this temporal delay corresponds to the time it takes the mean wind to transport the frozen turbulence field along the separation distance. The approach assumes the frozen turbulence hypothesis. This assumption makes it possible to measure one turbulent structure at different points in space when the separation between the points is aligned with the mean wind direction and when the time between the measurements equals the time it takes the mean wind to transport the turbulent structure from one point to the other. The line-of-sight measurements taken by the Windcube are unfortunately not continuous. Therefore, the chosen temporal delay can only be a multiple $n$ of the refresh rate of a particular line-of-sight measurement, i.e., $\tau = n \cdot 3.85\,\mathrm{s}$. As a consequence, the actual longitudinal separation distances for a squeezed pair of radial velocity measurements cannot become zero. But geometrical considerations show that they are reduced to

$$r_{\mathrm{real,SQZ},ij} = \Delta t_{ij} U.$$

An example is given in Fig. 2 where the lengths of $r_{\mathrm{real},ij}$ can be compared with the lengths of $r_{\mathrm{real,SQZ},ij}$. This shows that it is impossible to completely avoid the resonance effect due to longitudinal separation. However, it is possible to shift the resonance wave number away from the high energy region into a lower energy region where the measurement signal is already strongly attenuated by the line-one-sight averaging. The lateral separations, on the contrary, remain unchanged by the application of squeezed processing.

## 3 Methods

### 3.1 Field measurements

The measurement data used for this study originate from a measurement campaign in which a Windcube V2 was collocated to the $116.5\,\mathrm{m}$ high meteorological mast at the Danish National Test Center for Large Wind Turbines at Høvsøre, Denmark. The test location lies approximately $1.7\,\mathrm{km}$ east of the North Sea which is bordered by a stretch of dunes. Otherwise the terrain has no significant elevations. For reference measurements, the meteorological mast is equipped with Metek USA-1 ultrasonic anemometers at $10\,\mathrm{m}$, $20\,\mathrm{m}$, $40\,\mathrm{m}$, $60\,\mathrm{m}$, $80\,\mathrm{m}$, and $100\,\mathrm{m}$ heights. For a more detailed description of the test site we refer to Peña et al. (2016).

The measurements span a period from 11.09.2015 until 26.05.2016, with no measurements taken between 09.11.2015 and 17.02.2016. The lidar is positioned around $13\,\mathrm{m}$ to the west of the meteorological mast and oriented with its LOS1 into the north-east direction so that $\theta_0 = 45°$. An overview about the orientation of the lidar beams is given in Fig. 4.

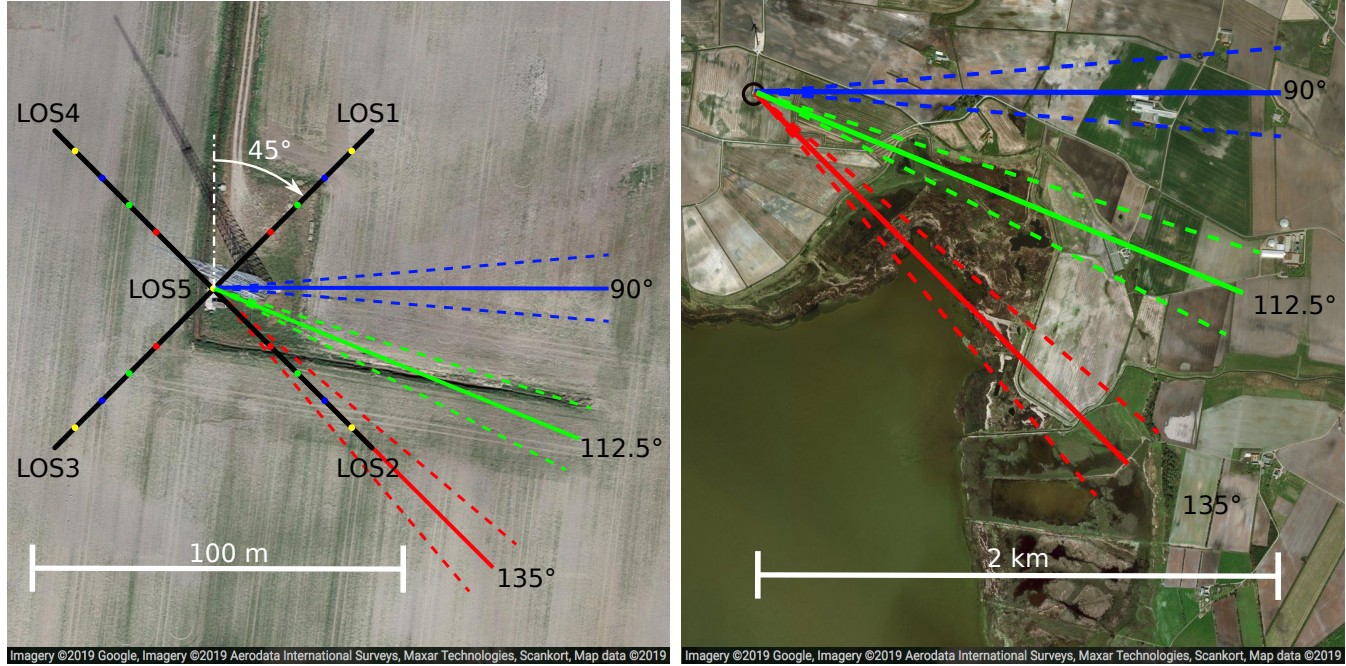

**Figure 4.** Aerial pictures of the location of the Windcube 13 m to the west of the meteorological mast at Høvsøre with the location of the measurement points along the lines-of-sight (left) and the landscape around the measurement location in the inflow directions (right). Top north. Adapted from Google Maps.

## 3.2 Sampling in a turbulence box

Sampling in a turbulence box is a method to simulate wind lidar measurements in very large computer-generated wind fields. The creation of such wind fields according to Mann (1998) requires less computational power than for example large eddy simulations (LES). LES was successfully used before to analyse coherent structures in wind fields (e.g. Stawiarski et al.,

2015) and wind profiles (e.g. Gasch et al., 2019) but predicting lidar derived turbulence velocity spectra requires much more turbulence data. An advantage of using LES is that Taylor's frozen turbulence hypothesis does not need to be applied but a drawback is that fine scale turbulence would be suppressed.

To be able to predict lidar derived spectra in a turbulence box, we first determined the three model parameters, i.e. the turbulence length scale $L$, the degree of anisotropy $\Gamma$, and the dissipation factor $\alpha \epsilon^{2/3}$ for all test cases by fitting the sonic

derived spectra to the Mann (1994) uniform shear model of turbulence. We used these parameters then to create large turbulence files that contain possible values of the three velocity components $u$, $v$, and $w$. In order to limit the required memory, we divided the desired box size into 32 separate files with different random seeds for each test case. Each of the files consists of 32 768 $\times$ 128 $\times$ 32 points. The selected spatial resolution is 2 m per point so that all files for one test case represent an air volume of 2 097 152 m length, 256 m width and 64 m height. These boxes contain turbulence statistics that are similar to what the

underlying spectral tensor describes. We created a Matlab script that samples data within the turbulence boxes similar to how

a Windcube samples wind velocities in the real atmosphere. The script first imports the turbulence files and cuts them into ten-minute intervals whose spatial length depends on the desired mean wind speed $U$. The script then considers a realistic timing by importing the timestamp data of an arbitrary Windcube .rtd file, which is a standard output data file type that contains the line-of-sight velocities of every single beam including their timing and carrier-to-noise ratio. Next, it defines the location of
the center of the range gate for all beams at all desired height levels within a ten-minute interval. Different inflow directions are imitated by altering the orientation of the beams with $\theta_0$. These locations are then moved into the horizontal central plain of the turbulence box. Centered around the midpoints of the range gates the program defines a total of 27 points along all lines-of-sight. These points have a distance of $1\,\mathrm{m}$ from each other. The turbulence velocities are then interpolated to these 27 points and projected onto the line-of-sight direction. A triangular weighting function is eventually multiplied to calculate
the line-of-sight averaged radial velocities. From this point on, the data processing is identical to the processing of the lidar measurement data as described in subsection 2.3.

## 3.3  Data selection

We filter the field data to include only the ten-minute intervals in which the mean wind velocity at $80\,\mathrm{m}$ above the ground was within an interval of $U = 8 \pm 0.5\mathrm{ms}^{-1}$. The reference height of $80\,\mathrm{m}$ was selected arbitrarily. Using only one reference height
in the filtering process assures that the same ten-minute intervals are used for all four investigated height levels: $h_1 = 40\,\mathrm{m}$, $h_2 = 60\,\mathrm{m}$, $h_3 = 80\,\mathrm{m}$ and $h_4 = 100\,\mathrm{m}$. The mean wind velocity $U = 8\mathrm{ms}^{-1}$ was selected because it is the most frequent in the data set. A narrow velocity bin is selected, so that the time delay used in the processing of actual measurements is identical with the time delay chosen for sampling in a turbulence box. Three narrow wind sectors around $\overline{\Theta}_1 = 135°$, $\overline{\Theta}_2 = 112.5°$ and $\overline{\Theta}_3 = 90°$ are chosen for the analysis. The width of the sectors is $\pm 5°$. In the first case, the wind is aligned with two of the
lines-of-sight, namely LOS2 and LOS4 ($\alpha = 90°$), in the second case the offset is $22.5°$ ($\alpha = 67.5°$) and in the third case the offset is $45°$ ($\alpha = 45°$). As shown in figure 4, the three inflow directions are dominated by flat farm land and the water of Nissum Fjord. The small town of Bøvlingbjerg lies in the east-south-east direction and is approximately $3\,\mathrm{km}$ away. Within $2\,\mathrm{km}$, only one farm might have some minor influence on the measurements in the first wind sector. The selected measurement sectors are neither affected by the wind turbines to the north, nor by the sea-to-land transition to the west of Høvsøre. The data
is in addition filtered to only contain intervals of neutrally stratified atmospheric conditions in order to achieve a good fit with the Mann model of turbulence. The filter criterion is a Monin-Obukhov length $|L_{MO}| > 500\,\mathrm{m}$ based on measurements at $20\,\mathrm{m}$ above the ground. Furthermore, to assure high quality of the analyzed measurement data, we filter out intervals with less than 100% data availability. Therefore, each line-of-sight measurement in the filtered dataset has a carrier-to-noise ratio better than the Windcube's standard threshold of -23 dB. After filtering, 49, 31, and 27 ten-minute intervals remain for the analysis of the
first, second and third wind sector, respectively.

## 3.4  Data processing

The lidar data from field measurements and sampling in a turbulence box are processed according to Eqs. 8 to 13. For every line-of-sight measurement, this processing creates a new component of the $\boldsymbol{u}_{\mathrm{DBS}}$ and the $\boldsymbol{u}_{\mathrm{SQZ}}$ vectors, where the subscript $_{\mathrm{SQZ}}$

indicates the squeezed wind vector reconstruction. In Fig. 2 two numbers are assigned to most of the measurement locations. The first number is increasing with the time of measurement. The second number though is increasing with the location along the mean wind direction. Where only one number is shown both numbers would be identical. In the process of reconstructing the squeezed wind vectors, it is essential to assign new timestamps that follow the order of the second numbers according

to where the measurements where taken. In practice, we project all measurement locations onto a vector that is pointing into the mean wind direction and evaluate all line-of-sight velocities in the order they fall along this vector. For reconstructing the horizontal wind speed components with the method of squeezing, we combine every radial velocity with the closest radial velocity originating from a beam with the opposite azimuth angle and being taken behind the current measurement location. The timestamp of this reconstructed component then depends on the average position of both measurement locations on the

mean wind vector. In order to create equidistant timestamps for the wind vectors $u_{\mathbf{DBS}}$ and $u_{\mathbf{SQZ}}$, we generate a linearly spaced time axis with $\Delta t = 0.96s$ and assign the wind components with the nearest neighbor method. This time step equals one quarter of the Windcube's cycle time and was chosen because the Windcube generates four wind vectors during one measurement cycle. Thus, we reach that all measurement data is used with no change in velocity variance which would occur if interpolation would be applied. The data from the ultrasonic anemometers is uniformly spaced with a sample rate of $20\,\mathrm{Hz}$

and is resampled to a rate of $4\,\mathrm{Hz}$ with an anti-aliasing filter applied to reduce the amount of data.

We calculate double-sided power spectral densities as functions of the wave number $k_1$

$$F_{ij}(k_1) = \frac{\langle \hat{u}_i \hat{u}_j^* \rangle}{N k_s} \tag{25}$$

where $\hat{\phantom{x}}$ is the discrete Fourier transformation, $^*$ the complex conjugate, $\langle \rangle$ the ensemble average of all ten minute intervals, $N$ the number of measurements in one interval, and $k_s = \frac{2\pi f_s}{U}$ is the sampling wave number, where $f_s$ is the sampling frequency.

For the cross-spectra $(i \neq j)$ we use the real part of $F_{ij}$. We then divide the $k_1$-axis into 35 logarithmically spaced bins and average the spectral values in each bin. By doing so we even out the spectra in the low wave number region, avoid the high density of data points in the high wave number region, and align the sonic and lidar values for ease of comparison. The spectral values are eventually pre-multiplied with their wave numbers and plotted on a linear vertical axis while the wave numbers are on a logarithmic horizontal axis. Displayed like this, any portion of the area under the spectra for a range of wave numbers is

proportional to the variance of the signal in this wave number range (Stull, 1988).

## 4   Results

Complete results are presented in the appendices A1 to A3. Here, we will present the results of two measurement height levels $h_2 = 60\,\mathrm{m}$ and $h_4 = 100\,\mathrm{m}$ and two inflow wind directions $\overline{\Theta} = 135°$ and $\overline{\Theta} = 90°$. These four cases alone show all relevant effects.

## 4.1 Simulation results

For the presentation of the results of our study, we will first discuss the simulated spectra without considering the experimental results. The lidar simulator opens up the possibility of analyzing the influence of the single wind velocity components on the spectra by switching them on or off in the turbulence box. This method helps in understanding what the final lidar spectra consist of. Figs. 5 and 6 show these simulated spectra for the inflow wind directions $\overline{\Theta} = 135°$ and $\overline{\Theta} = 90°$ respectively. The black solid lines are the target spectra that originate from sampling single points along the $u$-direction of the turbulence box with a frequency of 4 Hz. These target spectra are not completely smooth due to the finite length of the generated turbulence files, but they resemble the model spectra well enough for the purpose of this study. The red and yellow lines show the shape of the lidar spectra with conventional DBS processing and squeezed SQZ processing respectively. Solid lines are the resulting spectra when all three wind velocity components are switched on. Dashed lines show the spectra when only the $u$-component is activated. Dash-dotted lines represent spectra generated from the $v$-component alone and dotted lines are for the $w$-component alone. The method of showing the influence of the single components on the resulting lidar spectra cannot be used for cross-spectra. That is why we do not discuss the $uw$-spectra here but only show the results together with the measurements in subsection 4.2.

### 4.1.1 Aligned inflow

To begin with, we take a look at the results from $\overline{\Theta} = 135°$ inflow, i.e., the wind field is moving parallel to the azimuth angle of LOS2 and LOS4 (see Fig. 4). We see in Fig. 5 that only the $u$ and $w$ components of the wind field are involved in creating the lidar spectra of the $u$-component. With the method of DBS applied, the resulting lidar spectrum is correct only for very low wave numbers where $k_1 < 4 \times 10^{-3}\,\mathrm{m}^{-1}$. At increasing wave numbers the lidar underestimates the $u$-fluctuations in the wind field more and more, until it hardly detects them at the first resonance wave number, which is marked with a grey dashed vertical line. In parallel, the $w$-fluctuations contaminate the lidar measurements increasingly. Between the first and the second resonance wave number, the cross-contamination effect is lower again but it does not disappear completely. The reason is that two different longitudinal separation distances are involved in the wind vector reconstruction process as described at the end of subsection 2.4 ($r_{\mathrm{real}} \neq r_{\mathrm{rep}}$). We also see that the energy content at the second resonance wave number is much lower than at the first resonance wave number, although the $w$-fluctuations in the target spectrum in this wave number region are similarly strong. The reason is that the line-of-sight averaging is stronger for higher wave numbers and limits how much of the turbulence in the signal is being detected. The main difference between the two elevation levels 60 m and 100 m is that the resonance peaks are higher and shifted to the left for measurements at 100 m. The reason is mostly that the longer longitudinal separation distance at higher elevations corresponds to lower resonance wave numbers according to Table 2 and less line-of-sight averaging comes into effect at these lower wave numbers. The slightly different parameters of the underlying spectral tensors do of course also influence the results.

The wave number that corresponds to the sampling frequency of each lidar beam is marked with a grey solid vertical line. We cannot detect any turbulence at this wave number and the signal is strongly weakened close to it. This effect accounts for

$$\overline{\Theta} = 135°, \ \alpha = 90°$$

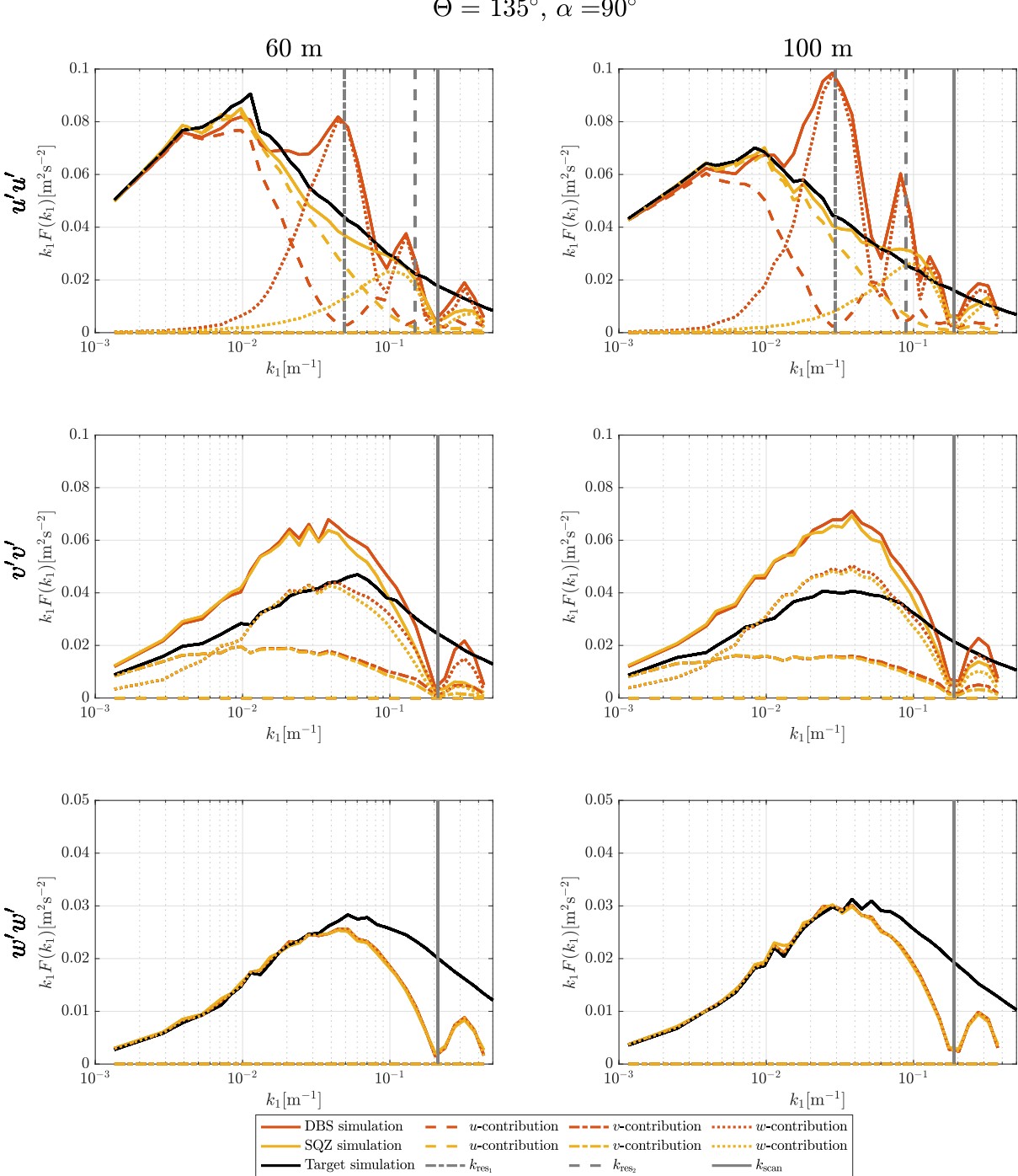

**Figure 5.** Turbulence velocity auto-spectra derived from sampling in a turbulence box for the case of aligned inflow with $\overline{\Theta} = 135°$ and $\theta_0 = 45°$. The measurement heights are $h_2 = 60\,\mathrm{m}$ (left) and $h_4 = 100\,\mathrm{m}$ (right). Black, red and yellow lines are target, DBS processed and SQZ processed lidar spectra. Dashed, dash-dotted and dotted lines show the influence of the $u$, $v$ and $w$-component on the resulting spectra. The vertical solid line marks the wave number that corresponds to the lidar sampling frequency $k_{\mathrm{scan}}$ and the vertical dashed lines show the first and second resonance wave numbers $k_{\mathrm{res}}$.

all test cases, wind velocity components, and elevations. For even higher wave numbers the measurement signal recovers, until the lidar spectra stop at the wave number that corresponds to half of the wind vector reconstruction frequency.

Comparing the results from conventional DBS processing with the results for squeezed processed SQZ sampling shows the striking advantage of the new method for aligned wind cases. The method of squeezing leads to $u$-spectra that are very similar to the target spectra. The region of the spectra that contains most of its kinetic energy is hardly contaminated. That is advantageous for example when the turbulence length scale is determined. The resonance point is shifted into the region where line-of-sight averaging and the attenuation due to the limited sampling frequency are strong. In the transition zone, the increasing averaging effect compensates for the increasing contamination. That means the very good agreement between target and lidar spectra is partly misleading and should not be interpreted as a perfect spectrum of pure $u$-fluctuations.

The situation is very different for the $v$-spectra. The conventional DBS processing hardly deviates from the squeezed processing. The small differences visible between the red and the yellow curves are due to the modified time scalar that is used in squeezed processing according to the description in the first paragraph of subsection 3.4. The lidar measured $v$-spectra contain the correct amount of spectral energy from the $v$-fluctuations only in the very low wave number region. As the coherence of the $v$-fluctuations declines at higher wave numbers, they become less detectable by the lidar. In addition, the lidar derived $v$-spectra are dominated by uncorrelated $w$-fluctuations due to the lateral separation of the involved measurement volumes. The squeezed processing does not improve the situation because it cannot decrease lateral separations.

The simulated spectra of the vertical wind velocity fluctuations $w$ are not contaminated by other wind speed components. The line-of-sight averaging becomes relevant for wave number of approximately $k_1 > 3 \times 10^{-2}\,\mathrm{m}^{-1}$. The strongest deviation from the target spectrum is found at the wave number $k_{\mathrm{scan}}$ that corresponds to the sampling frequency of the Windcube.

### 4.1.2 Non-aligned inflow

The situation is more complex for cases in which the incoming wind is not aligned with two of the lidar beams. As an example, we take a closer look at Fig. 6 that shows the simulation results for wind from $90°$. The inflow in this case is centered between two neighboring beams, which can be seen as the strongest case of non-aligned inflow. The behavior of all other inflow angles lies between this case and the previously discussed case of aligned wind from $135°$.

Even at the lowest wave numbers the estimation of the $u$-component is not correct. This is the most problematic characteristic of non-aligned inflow. From Table 3, we know that even without resonance, we cannot measure the $u$-component of turbulence correctly, if the lateral correlation is below unity. The spectra show that we indeed measure lower values of kinetic energy at low wave numbers by underestimating the $u$-fluctuations in the turbulence box. The contribution of $u$-fluctuations at increasing wave numbers becomes further reduced by the influence of the longitudinal resonance. Towards the resonance wave number contamination occurs. In addition to the contamination by the $w$-component like in the aligned wind case, we are also faced with some contamination from $v$-fluctuations. Due to the shorter longitudinal separations listed in Table 2 compared to the aligned wind case, the second resonance point is weakly pronounced, especially at $60\,\mathrm{m}$ elevation. The application of squeezed processing shifts the cross-contamination successfully into a region of lower energy content, but it cannot help derive better estimates of the turbulent energy in the low wave number region.

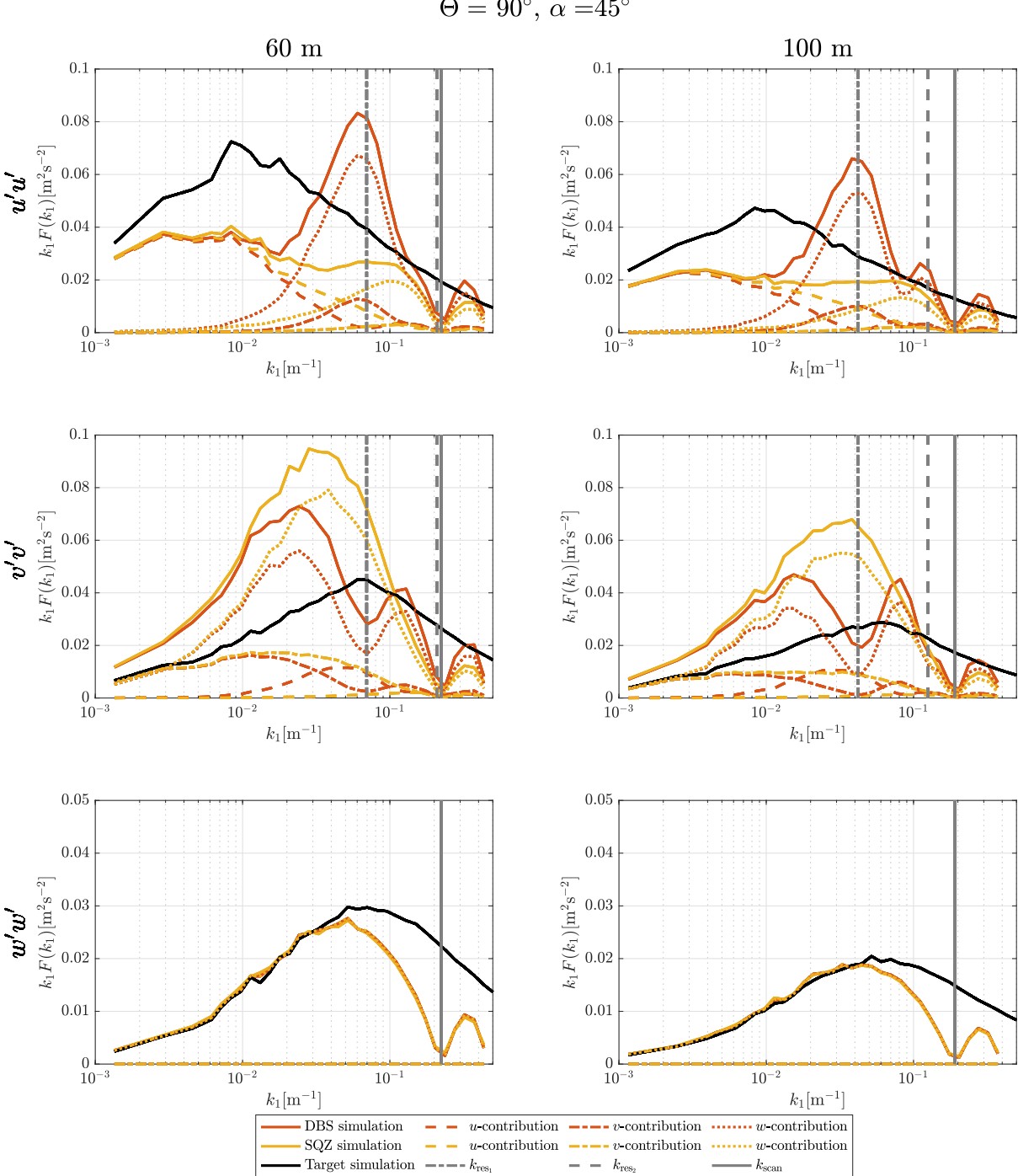

**Figure 6.** Turbulence velocity auto-spectra derived from sampling in a turbulence box for the case of non-aligned inflow with $\overline{\Theta} = 90°$ and $\theta_0 = 45°$. The measurement heights are $h_2 = 60\,\mathrm{m}$ (left) and $h_4 = 100\,\mathrm{m}$ (right). Black, red and yellow lines are target, DBS processed and SQZ processed lidar spectra. Dashed, dash-dotted and dotted lines show the influence of the $u$, $v$ and $w$-component on the resulting spectra. The vertical solid line marks the wave number $k_{\mathrm{scan}}$ that corresponds to the lidar sampling frequency and the vertical dashed lines show the first and second resonance wave number $k_{\mathrm{res}}$.

We now look at the predicted spectra of the transversal wind component $v$. In the very low wave number region, the actual $v$-fluctuations are nearly correctly interpreted due to the assumption of high lateral coherence of the $v$-component for very low values of $k_1$. Unfortunately, the spectra are contaminated by a significant parasitic contribution of $w$-fluctuations for which the coherence in the spectral tensor model is lower. With increasing decorrelation of the three wind velocity components at increasing wave numbers, the contamination becomes rapidly stronger. At the first resonance point, the cross-contamination of $v$ by $w$ is reduced but is to some degree replaced by cross-contamination from $u$-fluctuations.

The decreasing influence of $w$ and the additional cross-contamination by $u$ on the DBS lidar derived $v$-spectra can be removed by applying the method of squeezing. Nonetheless, the cross-contamination effect due to lateral separation is so strong that the spectra are not significantly better than the conventionally acquired ones. The DBS lidar derived velocity spectra for non-aligned wind are thus of limited use as they do not represent the actual wind conditions.

## 4.2 Comparison with measurements

Figs. 7 and 8 show the spectra for the same test cases as discussed in the subsection above. Now we compare the simulation results with measurement values. Markers in the plots are the spectra resulting from the field measurements, while solid lines, as before, correspond to the results from sampling in a turbulence box. First, we take a look at how well the theoretical target spectra displayed as black solid lines represent the spectra derived from the measurements of the sonic anemometers, which are depicted as black markers. The fitting of measurement data to the Mann spectral tensor model was successful. Overall, the model represents the measurements to a satisfactory degree. The measurement spectra show more scatter in the low wave number region which is random variation caused by the limited amount of analysed measurement data for the corresponding test cases. The agreement in the high wave number region where high statistical significance smooths out the derived spectra is in most cases very accurate. Discrepancies between sonic measurements and the spectral tensor in a certain wave number range have an effect on how well the theoretical spectra predict the lidar measurements. For example, the $v$ target spectra at both heights and wind directions show lower values for medium wave numbers than the measured spectra. The $uw$-target spectra, by contrast, show higher energy values in the low wave number region than what we actually measured. This has previously been reported by Mann (1994, Fig. 7a) and in Held and Mann (2019, Fig. C1). The uniform shear plus blocking (US+B) model by Mann (1994) and the model by de Maré and Mann (2016) match observations of the $uw$-spectrum better than the uniform shear (US) model of Mann (1994) that was used here, but they are much harder to implement and perform calculations with.

The method of sampling in a turbulence box is successful at predicting the shape of velocity spectra from a DBS scanning wind lidar. All characteristic features, i.e., cross-contamination, line-of-sight averaging, and limited sampling frequencies are found in the spectra of both measurements and simulations. But some deviations must be pointed out. In the test cases with non-aligned inflow from 90° and most other cases (Figs. A1-A3), the measured DBS processed $u$-spectra show increased values at wave numbers below the first interference wave number. That means that cross-contamination is likely stronger than predicted by the model at wave numbers below the first resonance point. We see three possible explanations for this behavior. First, Table 3 shows that the cross-contamination of the $u$-component by $w$-fluctuations for non-aligned wind inflow in the resonance case is much stronger when the coherence is high. Eliassen and Obhrai (2016) show for an offshore location and a

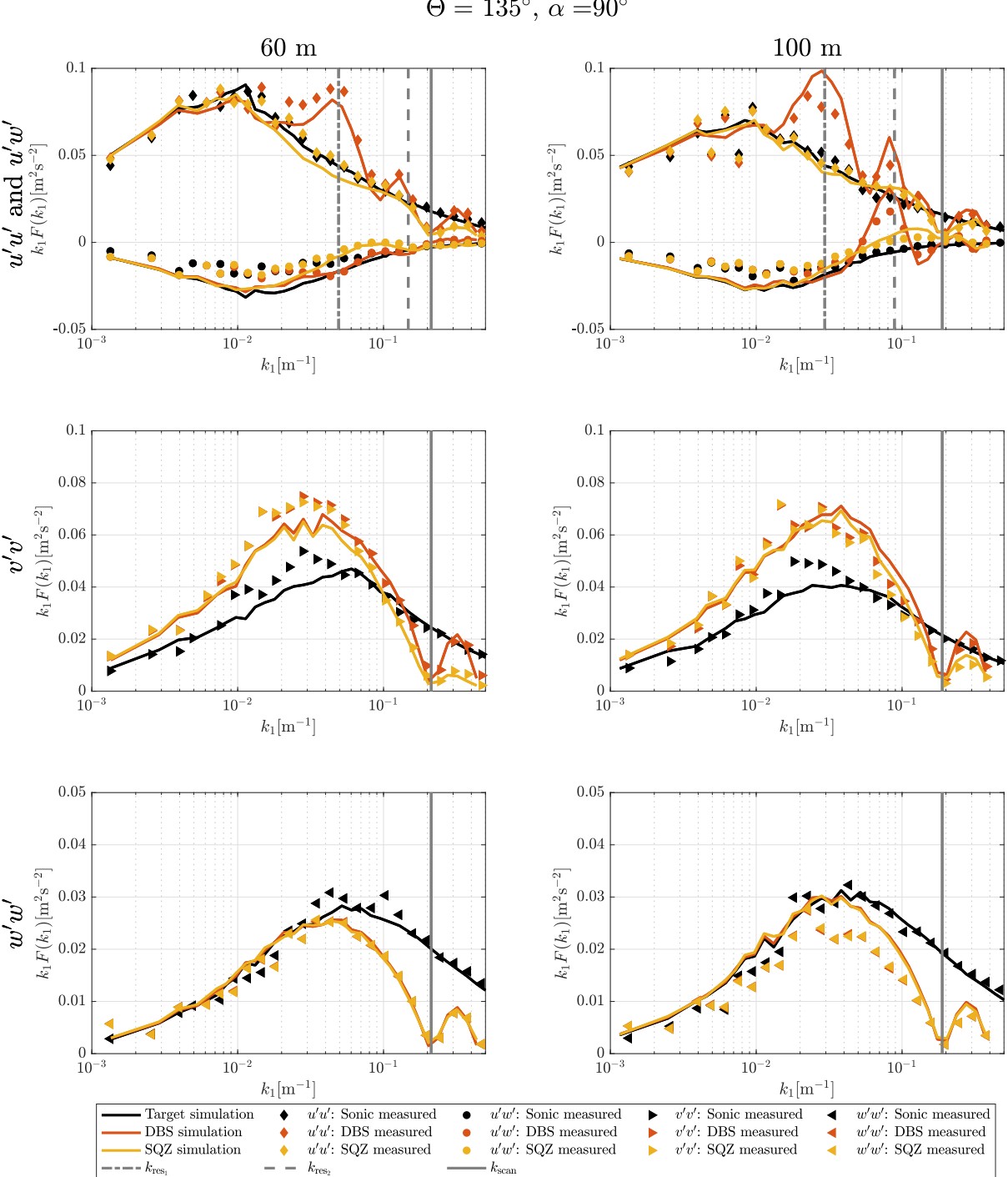

**Figure 7.** Turbulence velocity auto-spectra and $uw$-cross-spectra derived from sampling in a turbulence box and measurements for the case of aligned inflow with $\overline{\Theta} = 135°$ and $\theta_0 = 45°$. The measurement heights are $h_2 = 60\,\mathrm{m}$ (left) and $h_4 = 100\,\mathrm{m}$ (right). Black, red and yellow lines are target, DBS processed and SQZ processed lidar spectra from sampling in a turbulence box. Markers are spectra from field measurements. The vertical solid line marks the wave number that corresponds to the lidar sampling frequency and the vertical dashed lines show the first and second resonance wave number.

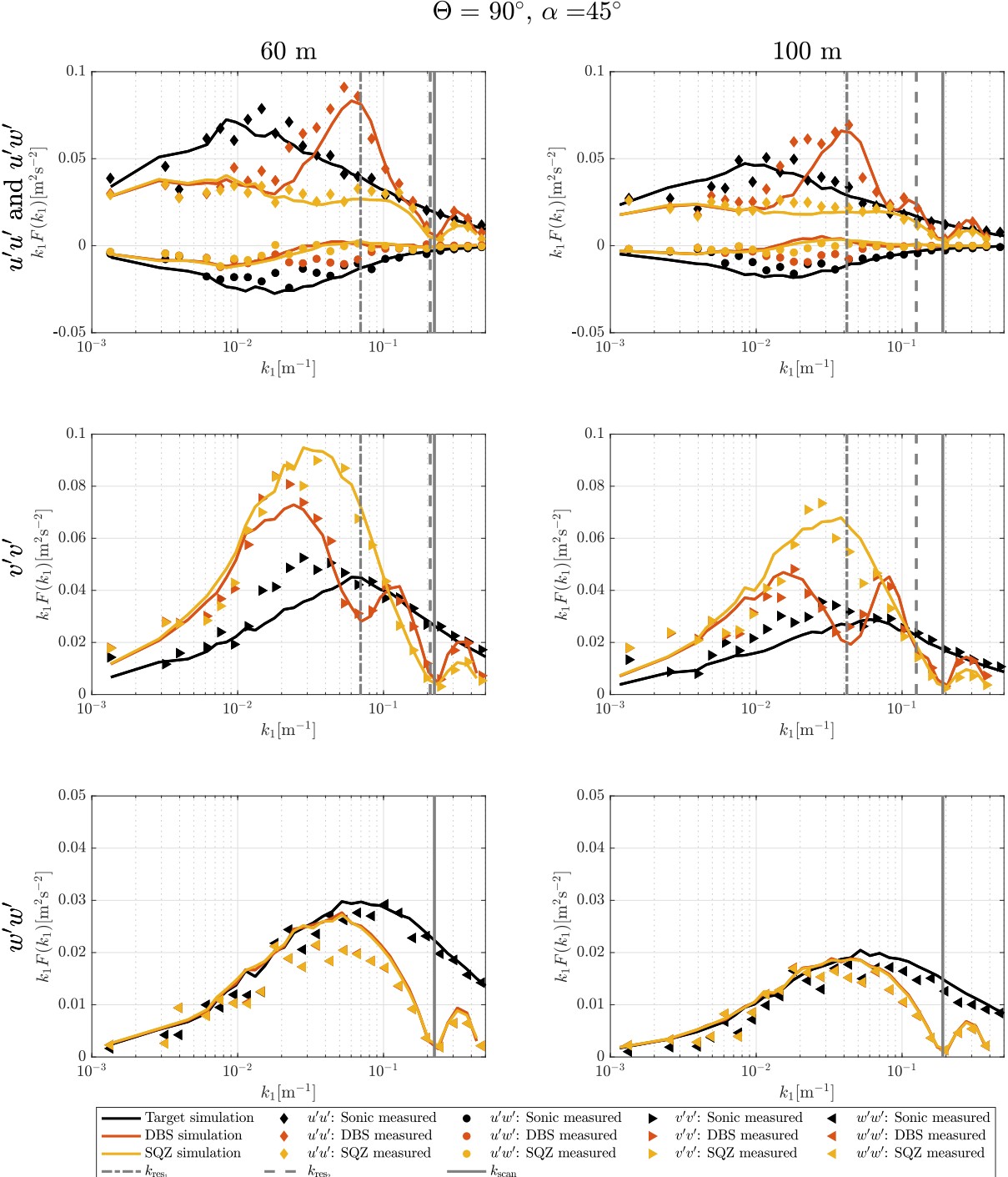

**Figure 8.** Turbulence velocity auto-spectra and $uw$-cross-spectra derived from sampling in a turbulence box and measurements for the case of non-aligned inflow with $\overline{\Theta} = 90°$ and $\theta_0 = 45°$. The measurement heights are $h_2 = 60\,\mathrm{m}$ (left) and $h_4 = 100\,\mathrm{m}$ (right). Black, red and yellow lines are target, DBS processed and SQZ processed lidar spectra from sampling in a turbulence box. Markers are spectra from field measurements. The vertical solid line marks the wave number $k_\mathrm{scan}$ that corresponds to the lidar sampling frequency and the vertical dashed lines show the first and second resonance wave number $k_\mathrm{res}$.

vertical separation of 40 m that the Mann model of turbulence underestimates the amount of coherence of the $w$-component in a wide range of wave numbers (see also Mann, 1994, Fig. 8). Assuming that the same occurs with transversal separations, we found a potential explanation for why the simulations of the non-aligned cases underestimate the $u$-variance at wave numbers below the resonance point. At higher wave numbers, the prediction is correct again because the correlation is close to zero, both in the spectral tensor and in reality. A second possible explanation lies in the limited validity of the frozen turbulence assumption. Real turbulence is not perfectly correlated over long separation distances, so uncorrelated $w$-fluctuations might contaminate the $u$-measurements. And third, we must also expect that turbulence is not always advected with the ten-minute mean wind speed $U$ but sometimes slower or faster. This influences at which wave numbers the cross-contamination occurs.

The prediction of the $u$-spectra resulting from squeezed processing is overall precise but has a slight tendency towards underestimating the spectral values in the medium wave number range. Based on the available data, it is not possible to determine the definite cause of the higher spectral values in the DBS and SQZ processed $u$ measurements. However, we assume that the main reason is inaccurate representation of the co-coherences in the wind by the chosen spectral tensor. Sathe et al. (2011) also predict slightly lower total $u$-variances and significantly lower $v$-variances with their model than they get from measurements. However, our predictions of $v$-variances are more accurate, and we therefore cannot draw conclusions from the comparison with their work.

The shape of the lidar derived spectra of the transversal component $v$ for both processing methods is fairly accurately predicted by the simulation. The few significant differences can in most cases be explained by the aforementioned discrepancies between the spectral tensor and the actual wind conditions. For example, at 135° at 60 m elevation, the lidar measured $v$-fluctuations in the wave number range around $k = 2 \times 10^{-2} \, \text{m}^{-1}$ are considerably stronger than predicted because the actual wind fluctuations in the $v$ and $w$ directions are also higher than assumed by the selected spectral tensor.

The spectra of the vertical wind fluctuations $w$ are in some cases very accurately predicted by the simulations, for example in the case with inflow from 135° at 60 m elevation. In other cases, we predict considerably higher values than what is measured, e.g., at 135° at 100 m elevation and vice versa for example at 112.5° at 80 m where we measure stronger low frequency turbulence with the lidar than with the sonic anemometer (Fig. A2). The reason for this behavior is unknown.

The $uw$-cross-spectra are predicted well for both data processing methods for aligned inflow. For inflow conditions in which the wind direction is not aligned with two of the beams, the prediction of the DBS processed data is off. We assume that the reason for this behavior is the same as what caused the differences between the DBS processed $u$ measurements and simulations.

## 5 Conclusions

We have shown that with the help of sampling in a turbulence box, it is possible to predict turbulence velocity spectra from DBS wind lidar for all wind directions. We have analyzed these spectra theoretically as well as in comparison with field measurements.

The shape of the spectra from a Windcube V2 DBS lidar is influenced by the effects of line-of-sight averaging, its limited sampling frequency, and strongly by cross-contamination. We have shown that the influence of cross-contamination on the spectra of the horizontal components of turbulence is dependent on the alignment of the lidar beams to the incoming wind direction. Only the measurement of vertical wind fluctuations is independent of wind direction due to the availability of a beam pointing vertically upwards. The auto-spectrum of each horizontal wind speed component is distorted by the influence of the other two wind components. Also the $uw$-cross-spectrum suffers from cross-contamination.

The method of squeezing applied in the wind vector reconstruction process minimizes the cross-contamination effect on the measured $u$-component of turbulence when the wind blows parallel to one of the beam's azimuth angles. Only in this case are the lidar derived spectra reasonably close to the spectra of the $u$-component of the wind, so that turbulence parameters like turbulence length scale and the dissipation factor might be estimated from it.

In all other cases, the estimations of the horizontal component spectra of turbulence are very erroneous due to the parasitic influence of the components of turbulence on one another and one should not trust them. In no case should turbulence velocity spectra from DBS wind lidar be fitted to a turbulence model.

Multi lidar arrangements use three separate lidar devices whose beams intersect at one point in space and minimize separation distances (Mann et al., 2009). A different possibility to avoid cross-contamination would be to deflect the inclined beams of one single DBS wind lidar first into a horizontal direction away from the device and second towards a point above the device where they intersect. Such a setup requires precise alignment of the deflected beams but would not require horizontal homogeneity of the wind field and could measure turbulence more accurately.

*Code and data availability.* All data and code used for this study can be downloaded from: https://doi.org/10.5281/zenodo.3514326

*Author contributions.* FK performed the data processing, analyzed the results, and wrote the paper. JM supplied the measurement data and gave input and advice throughout the process.

*Competing interests.* The authors declare that they have no conflict of interest.

*Acknowledgements.* This research project was supported by Energy and Sensor Systems (ENERSENSE) at the Norwegian University of Science and Technology. We thank Kam Sripada and Tania Bracchi for thorough proofreading of the manuscript.

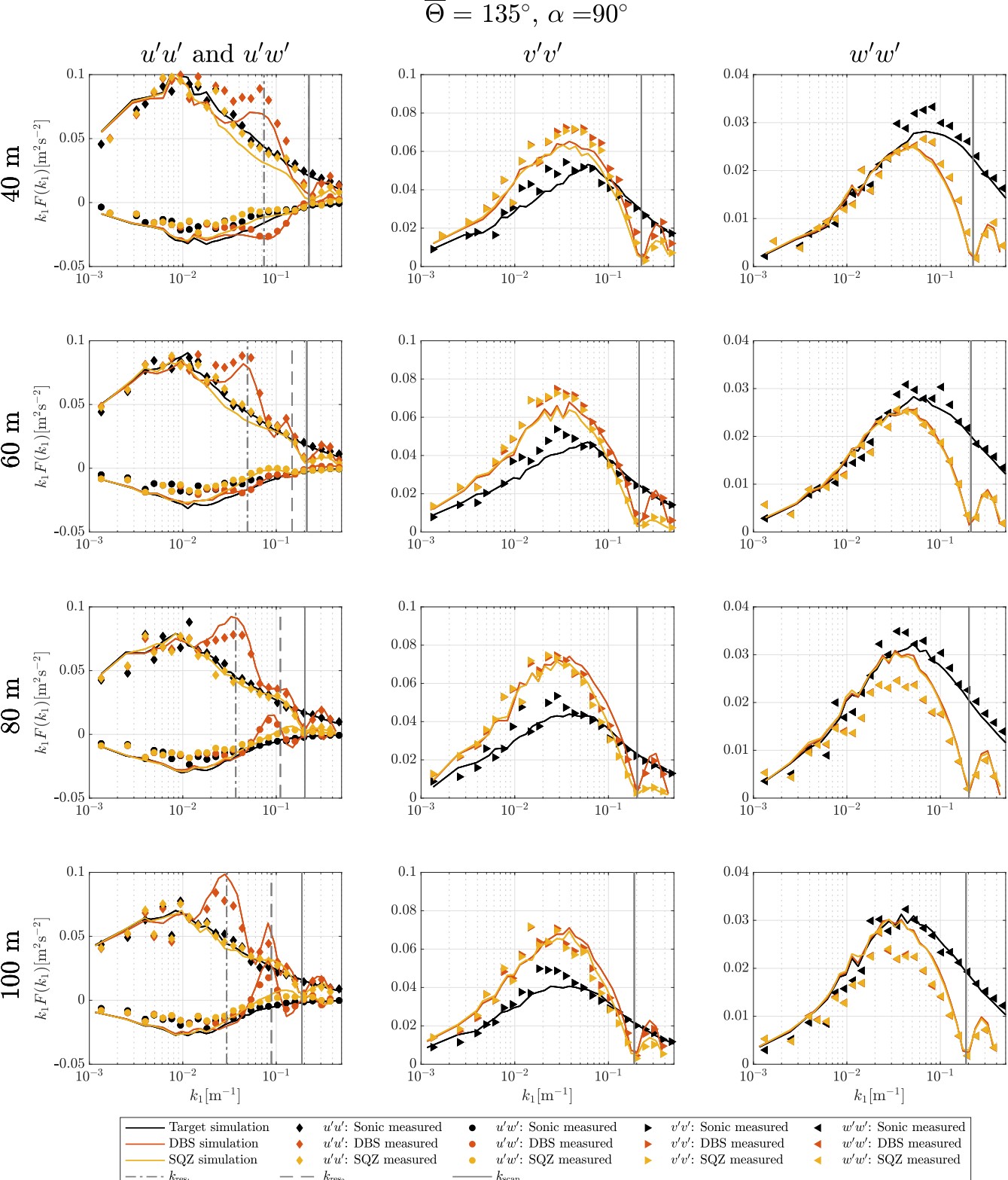

**Figure A1.** Turbulence velocity auto-spectra and $uw$-cross-spectra derived from sampling in a turbulence box and measurements for the case of aligned inflow with $\overline{\Theta}_1 = 135°$ and $\theta_0 = 45°$.

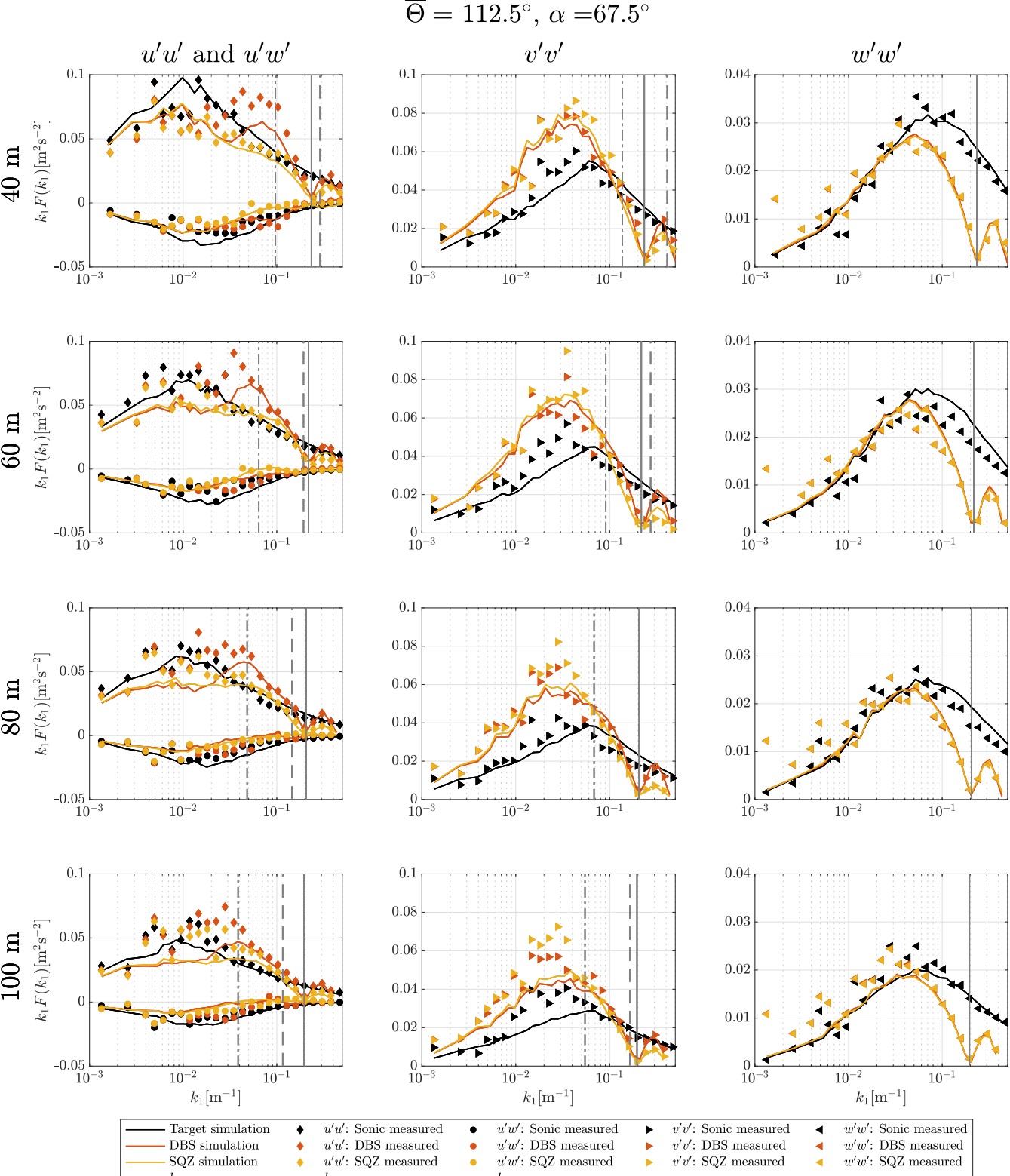

**Figure A2.** Turbulence velocity auto-spectra and $uw$-cross-spectra derived from sampling in a turbulence box and measurements for the case of non-aligned inflow with $\overline{\Theta}_2 = 112.5°$ and $\theta_0 = 45°$.

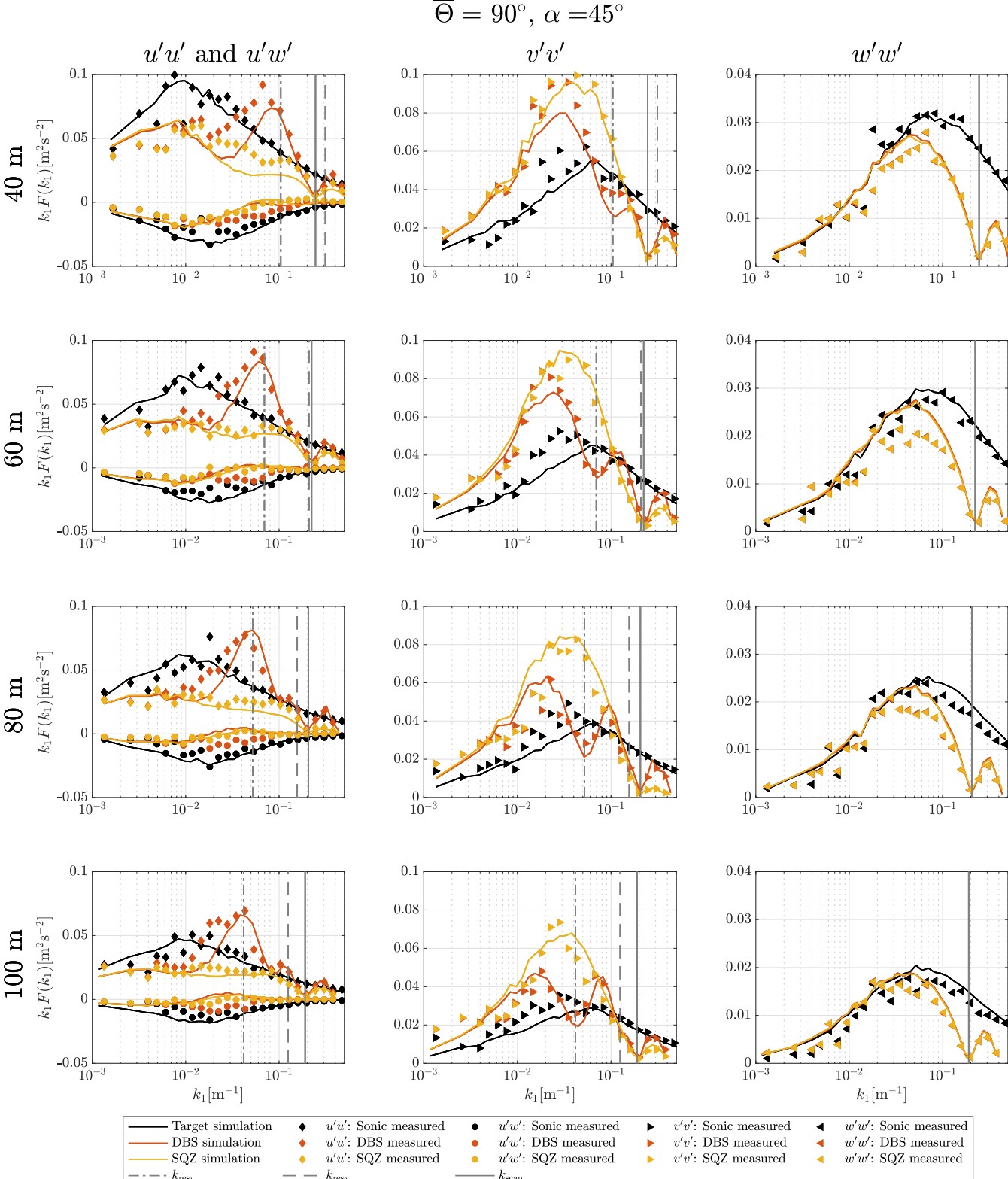

**Figure A3.** Turbulence velocity auto-spectra and $uw$-cross-spectra derived from sampling in a turbulence box and measurements for the case of non-aligned inflow with $\overline{\Theta}_3 = 90°$ and $\theta_0 = 45°$.

| | | |
|---|---|---|
| $c$ | [ms$^{-1}$] | Speed of light |
| $D$ | [m] | Diameter of measurement cone |
| $d_f$ | [m] | Distance from lidar to center of range gate |
| $F$ | [m$^2$s$^{-1}$] | Power spectral density |
| $f_s$ | [s$^{-1}$] | Sampling frequency |
| $h$ | [m] | Measurement height |
| $i, j$ | [] | Beam numbers 1...5; Wind vector components 1...3 |
| $k$ | [m$^{-1}$] | Wave number |
| $k_s$ | [m$^{-1}$] | Sampling wave number |
| $k_{\mathrm{res}}$ | [m$^{-1}$] | Resonance wave number |
| $k_{\mathrm{scan}}$ | [m$^{-1}$] | Wave number of LOS sampling frequency 0.26 Hz |
| $l_p$ | [m] | Half length of range gate |
| $N$ | [] | Number of measurements per 10-minute interval |
| $n$ | [] | Integer index |
| $\boldsymbol{n}_i$ | [] | Unit vector along beam $i$ |
| $r_{\mathrm{lat},ij}$ | [m] | Nominal separation distance in lateral direction w.r.t. $\overline{\Theta}$ for beam combination $ij$ |
| $r_{\mathrm{long},ij}$ | [m] | Nominal separation distance in longitudinal direction w.r.t. $\overline{\Theta}$ for beam combination $ij$ |
| $r_{\mathrm{rep},u}$ | [m] | Representative separation distance in longitudinal direction w.r.t. $\overline{\Theta}$ for the reconstruction of $u$ |
| $r_{\mathrm{rep},v}$ | [m] | Representative separation distance in longitudinal direction w.r.t. $\overline{\Theta}$ for the reconstruction of $v$ |
| $r_{\mathrm{real},ij}$ | [m] | Real separation distance in longitudinal direction w.r.t. $\overline{\Theta}$ for beam combination $ij$ considering $t$ |
| $r_{\mathrm{real,SQZ},ij}$ | [m] | Actual separation distance in longitudinal direction w.r.t. $\overline{\Theta}$ for beam combination $ij$ considering $t$, squeezed processing |
| $s$ | [m] | Distance from center of range gate |
| $t$ | [s] | Beam timing |
| $\boldsymbol{u}, \boldsymbol{U}, \boldsymbol{u}'$ | [ms$^{-1}$] | Total, mean, and fluctuating part of wind velocity vector |
| $u, v, w$ | [ms$^{-1}$] | Longitudinal, transversal, and vertical wind velocity component w.r.t. $\overline{\Theta}$ |
| $V_{\mathrm{hor}}, \overline{V}_{\mathrm{hor}}$ | [ms$^{-1}$] | Horizontal wind velocity, ten minute mean |
| $v_{r_i}$ | [ms$^{-1}$] | Radial wind velocity in beam $i$ direction |
| $\tilde{v}_{r_i}$ | [ms$^{-1}$] | Line-of-sight velocity of beam $i$ |
| $\boldsymbol{x}$ | [ms$^{-1}$] | Wind velocity vector in Windcube coordinates |
| $x, y, z$ | [ms$^{-1}$] | Wind velocity component in LOS1 – LOS3, LOS2 – LOS4, and LOS5 direction |
| $\alpha$ | [°] | Relative inflow angle $\overline{\Theta} - \theta_0$ |
| $\theta_0$ | [°] | Heading of LOS1 (offset from north) |
| $\theta$ | [°] | Beam azimuth angle |
| $\Theta, \overline{\Theta}$ | [°] | Wind direction, ten minute mean |
| $\sigma^2$ | [m$^2$s$^2$] | Velocity variance |
| $\phi$ | [°] | Zenith angle (half cone opening angle) |
| $\varphi$ | [] | Triangular weighting function |

**Table A1.** Nomenclature

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
