# Peer review of "Cross-contamination effect on turbulence spectra from Doppler beam swinging wind lidar"

_Wind Energy Science, 2019_

## Author Comment (AC1) · 20 Oct 2019

We invite interested readers to download the data and processing scripts used for the manuscript:

https://doi.org/10.5281/zenodo.3514326

---

## Referee Comment (RC1) · Anonymous Referee #1 · 25 Nov 2019

The article by Kelberlau and Mann addresses an interesting problem that profiling lidar measurements are facing. Now-a-days, the site suitability of wind plants depends on the profiling lidar data, and the cross-contamination issue is a great concern for it. Thank you for working on this topic.

The article provides the required background theories and explanations to follow the document. The article is a follow-up of their previous article (Kelberlau and Mann (2019)) that they have published on a technique called squeezed wind vector reconstruction to reduce the cross-contaminations of lidar measurements. This article is an application of the published technique to the WindCube V2 profiling lidar data. The published technique is based on the Taylor 'frozen' hypothesis (turbulent eddies are advected by the mean wind speed), and the hypothesis can only be applied along the mean wind direction. However, the authors have worked on the non-aligned (line-of-sight (LOS) is not aligned to the mean wind direction) flow using the concept of Taylor 'frozen' hypothesis. In the end, the authors show that their method works mainly for the aligned flow. Even for the aligned flow the technique does not work for the spanwise component. Therefore, the application and effectiveness of the work is limited.

In the article, the authors have introduced separation distance as a part of the squeezed reconstruction technique. Separation distance should a parameter based on the mean wind speed and time lag. The authors have mentioned that separation distance represents statistical average, not the actual separation, and I do not understand the exact application of this parameter in this work. The temporal frequency of the data collection by the lidar is low unlike continuous wave lidar (what the authors have worked on their previous article) and LOS measurements (full scan) are updated in every 3.85 s. The authors have not provided any clear data on the amount of reduced distance (and corresponding time delay) between the LOS measurements due to their technique. Considering the low temporal frequency of the lidar (WindCube V2) data, the authors will not get measurement data at the target time after the considered advection. The method would be beneficial if there is high frequency data so that the user can get more data in space to take the benefit of the reduced spatial distance. It is not clear what the authors do here. It would be nice if the authors provide a block diagram of the work process of the squeezed reconstruction method applied in this work (with a sample data). In addition, showing a figure like Figure-4 of Kelberlau and Mann (2019) article would be nice.

Coherence model based on mesurements is showing that the longitudinal coherence drops to approximately zero with 90 m separation distance and the corresponding wave number is close to 0.06 (Davoust et al. 2016). Then why is the Taylor 'frozen' hypothesis effective here?

Specific comments:
1. Page-4, L-8: These line-of-sight velocities are the product of…
   Is this the way lidar measures? We numerically model the lidar measurements in this way. Make it clear.

2. WindCube V2 has pulse repetition frequency (PRF) 30k. The default PRF used by the lidar is 20k; then required time for LOS measurements should be 30/20=0.6667 s. Why it is 0.72 s? And also, why the required total time to finish the scan pattern is 3.85 s?
3. How do the authors get Equation 17?
4. Are all the conditions provided in Page-9, L-10 and L-11 correct?
5. Is $K_{scan}$ (Equation-23) related to the scan pattern time?  Or is it particularly related to the vertical beam ($5^{th}$ beam)? Due to the vertical beam, there are no measurements for horizontal wind speed. Could the authors make a comment on it?
6. Page -11, L-19: what do the authors mean by, "for the combination of LOS2 and LOS4 it is similar"?
7. Do the authors use Equation 25 (separation distance, $r_{rep}$) to calculate the time delay? How Equation 25 is different from Equation 24? It is not clear here.
8. Second paragraph of Page 12, L-8 to L-12: "The methods of squeezing.." is irrelevant here. Remove the whole paragraph.
9. Page-14, L-6: In case of three?
10. Page-14, L-20 to L-25: Rewrite the sentences so that reader can understand the process: "In the process of reconstructing…"
11. Page-17, L-1: The authors mention to explain the results: "the reason is that two different longitudinal separation distances are involved in the wind vector reconstruction process…. ". However, in Table-1, authors showed that the second longitudinal separation distance is zero for the aligned flow. Could the authors explain what are they trying to explain here?
12. This article is particularly trying to solve an active industry problem and the article needs to be coherent in terms of explanation which I don't see in Section 4. Explanation of the results is so vague that I get always lost.

---

## Referee Comment (RC2) · Stefan Emeis (Referee) · 18 Dec 2019

This manuscript gives an important analysis on how reliable are turbulent spectra derived from wind lidar measurements. This helps with the derivation of the necessary information for planning and operating wind turbines.

Main idea of this manuscript is sampling turbulence data from a computer-generated turbulence box. This gives the perfect opportunity to vary all relevant parameters necessary for an reliable assessment.

All this is presented in a very clear and concise style. Thus, I recommend to publish

this manuscript as is.

---

## Referee Comment (RC3) · Anonymous Referee #3 · 28 Dec 2019

Kelberlau and Mann present work towards an improved measurement of turbulence spectra from Doppler lidar DBS scans. They introduce a methodology to simulate the lidar measurements in a turbulence box which helps them to analyze the quality of the lidar measurements. With the method of squeezing that has been introduced in a previous study they achieve remarkable improvements by eliminating cross-contamination effects in the lidar measurements. They show that these improvements can only be achieved if the wind speed is aligned with the DBS scan and conclude that in all other conditions, the spectra cannot be corrected. I think this study provides very interesting analysis and important insights into DBS scanning. However, I found the manuscript hard to read in some parts, mostly because of unprecise language and variable definition. Despite this there are some other major concerns which I summarize in the general comments. I recommend the manuscript to be considered for publication in Wind Energy Science after major revisions.

**0.1 General comments**

- I think the introduction can be improved to better motivate the use of DBS scans for turbulence retrieval. There are many studies that use VAD-scans for this purpose. What is the advantage of using DBS? Please relate this to the work of Eberhard, Frehlich, Smalikho, Krishnamurthy, Bodini etc.

- Since this is a manuscript for Wind Energy Science, I think the authors should describe a little bit more how turbulence spectra can be used in practice for wind energy purposes. I think many wind energy experts are not very familiar with this topic. How exactly do they relate to IEC 61400-1

- Please be very clear with directions, angle offsets and definitions. It is quite hard to follow the different coordinate systems that are used throughout the manuscript. A nomenclature of variables in the appendix would also help to serve this purpose.

- Section 4 with the results stops at describing the differences between measurements and simulation in a qualitative way. I want to encourage the authors to consider adding a quantification of the error between lidar and sonic estimated turbulence parameters at least for the cases with aligned wind flow with the DBS scan. Also, many unknown behaviours are described without giving ideas about how to investigate this behaviour any further. This could be added to the conclusion.

- The conclusion and outlook section is very short with a rather pessimistic ending stating that in most cases the turbulence spectra "should not be trusted". I think

these findings should be related to the goal of wind site assessment and load prediction that is mentioned in the beginning. What are the prospects? How can this work help in future? What are alternative measurements that could be done for this purpose and what are the advantages/disadvantages compared to the method presented in this study. One question that came to my mind is if a DBS strategy which adapts the beam direction to the wind direction could be used to overcome the problem of cross-contamination.

- I recommend some language copy-editing if the manuscript is accepted for publication.

**0.2 Specific comments**

- p.1, l.2f: The authors write that DBS lidars generate spectra. This is confusing, because it suggests that there is only one kind of velocity spectra and it is automatically produced by the lidar. I think the authors should be very clear from the beginning how these spectra are produced (i.e. from radial velocities, vertical stare or the retrieved wind vector).

- p.1, l.7: The method of squeezing should maybe be briefly introduced, because it is not a well-known term in the community.

- p.2, l.20ff: There exist some works that simulate lidar scans in LES fields (e.g. Stawiarski et al., 2015). What are differences / advantages of the method using the turbulence box. This could be described in more detail in Section 3.2.

- p.3, l.10: How is the time scale defined that divides the mean part from the turbulence part in the Reynolds decomposition?

- p.7, ll. 9ff: I cannot follow how Eq. 16 and 17 are concluded from Eq. 13, 8 and 9. Also, it is defined in Sect. 2.1 that $u$ is the longitudinal wind component and $v$

the transversal, but now it seems that these are the meteorological conventions!?

- p.8, Eqs.18-22: I think these equations could be presented in a more concise way for better readibility. For example, $\overline{\Theta} - \theta_0$ could easily be replaced by a single variable name and $\sigma_u^2$ in Eq. 21 could be presented as a function of $u_{DBS}$. By the way, $DBS$ as the variable subscript is a bit unfortunate. More than one letter in the subscript should not be italic.

- p.12, l.8: ZX300 was only briefly mentioned in the introduction. Maybe repeat here what is meant with the abbreviation.

- p.12, l.12: What are .rtd-files. The file ending is not really important for the reader, but what kind of information they contain!

- p.12, l.25: The parameters should be introduced with their meaning.

- p.14, l.22: "project all focus points onto a vector..." I think this is unclear. What are the focus points in a pulsed lidar?

- p.14, l.27: Is a nearest neighbour method really the best solution? Would interpolation not be better (even though it would definitely also not be perfect in a turbulent flow)?

- p.14, ll.31ff: I recommend putting this description in a mathematical formula.

- p.15, l.6f: That the area under the power spectral density must equal the variance of the time signal follows from the Parseval's theorem and should always be checked and valid if power spectra are calculated. However, with the scaling with the wave number as it is done in Fig. 4, this does not apply. Please check and add the relevant literature and formula, if you mention it.

- p.15, l.17: What is the "long axis"? Please be more specific. Also, define what is the "target spectra". Do these spectra contain volume averaging of the lidar?

- Fig.4 Fig.5: The line styles of $u$- and $v$-component are hard to distinguish. It would be good to show the $k^{-5/3}$-slope in the plots to get an idea of how well the spectra fit to the inertial subrange theory.

- p.28, l.14: I think the number of the IEC-standard should appear in the reference.

---

## Author Comment (AC4) · 23 Jan 2020

We thank Prof. Emeis for his very encouraging and friendly review.

---

## Author Response (AR1)

We thank Anonymous Referee #1 a lot for reviewing our manuscript thoroughly. Their feedback helped us to improve the text, so that it is easier to read and understand. Below you find a copy of the referee's comments and our response marked in red.

Anonymous Referee #1

The article by Kelberlau and Mann addresses an interesting problem that profiling lidar measurements are facing. Now-a-days, the site suitability of wind plants depends on the profiling lidar data, and the cross-contamination issue is a great concern for it. Thank you for working on this topic.

The article provides the required background theories and explanations to follow the document. The article is a follow-up of their previous article (Kelberlau and Mann (2019)) that they have published on a technique called squeezed wind vector reconstruction to reduce the cross-contaminations of lidar measurements. This article is an application of the published technique to the WindCube V2 profiling lidar data. The published technique is based on the Taylor 'frozen' hypothesis (turbulent eddies are advected by the mean wind speed), and the hypothesis can only be applied along the mean wind direction. However, the authors have worked on the non-aligned (line-of-sight(LOS)is not aligned to the mean wind direction) flow using the concept of Taylor 'frozen' hypothesis. In the end, the authors show that their method works mainly for he aligned flow. Even for the aligned flow the technique does not work for the spanwise component. Therefore, the application and effectiveness of the work is limited.

In the article, the authors have introduced separation distance as a part of the squeezed reconstruction technique. Separation distance should a parameter based on the mean wind speed and time lag. The authors have mentioned that separation distance represents statistical average, not the actual separation, and I do not understand the exact application of this parameter in this work. The temporal frequency of the data collection by the lidar is low unlike continuous wave lidar (what the authors have worked on their previous article) and LOS measurements (full scan) are updated in every 3.85 s. The authors have not provided any clear data on the amount of reduced distance (and corresponding time delay) between the LOS measurements due to their technique.

In section 2.5 we describe that the longitudinal separation distances are reduced to the values given by Eq. (25) when the method of squeezed processing is applied. A visualization of the conventional and reduced longitudinal separation distances is included in the revised version of the manuscript (Figure 2).

Considering the low temporal frequency of the lidar (WindCube V2) data, the authors will not get measurement data at the target time after the considered advection. The method would be beneficial if there is high frequency data so that the user can get more data in space to take the benefit of the reduced spatial distance. It is not clear what the authors do here. It would be nice if the authors provide a block diagram of the work process of the squeezed reconstruction method applied in this work (with a sample data). In addition, showing a figure like Figure-4 of Kelberlau and Mann (2019) article would be nice.

We added Figure 2 to the manuscript that we think adds clarity to what we do. It shows the position of measurement locations in parallel to Fig. 4 of Kelberlau and Mann (2019) but will also help to better understand the concept of longitudinal and lateral separation distances.

Coherence model based on measurements is showing that the longitudinal coherence drops to approximately zero with 90 m separation distance and the corresponding wave number is close to 0.06 (Davoust et al. 2016). Then why is the Taylor 'frozen' hypothesis effective here?

Davoust and Terzi (2016) measure low longitudinal coherences upstream of an operating wind turbine. In a similar experimental setup, e.g. Schlipf et al. (2015) find longitudinal coherences which are stronger than the values of Davoust and Terzi (2016) but still far from unity. Both studies measure in the induction zone of operating wind turbines which might have an influence on the

longitudinal coherence. Also yaw-misalignment might reduce the measured coherence. We measure in undisturbed flow where Taylor's frozen turbulence appears to be a reasonable assumption for the wave numbers of interest.

D. Schlipf, F. Haizmann, N. Cosack, T. Siebers and P. Wen Cheng, Detection of Wind Evolution and Lidar Trajectory Optimization for Lidar-Assisted Wind Turbine Control, Meteorologische Zeitschrift, Vol. 24, No. 6, 565–579, 2015

Specific comments:

1. Page-4, L-8: These line-of-sight velocities are the product of...Is this the way lidar measures? We numerically model the lidar measurements in this way. Make it clear.

In order to make clear what the lidar does and how we model this behaviour we changed this passage to: *"These line-of-sight velocities are the weighted average of the radial wind velocities along the stretch of the lidar beam illuminated by the range gate. A reasonable weighting function to model the line-of-sight averaging is the convolution…"*

2. WindCube V2 has pulse repetition frequency (PRF) 30k. The default PRF used by the lidar is 20k; then required time for LOS measurements should be 30/20=0.6667s. Why it is 0.72 s? And also, why the required total time to finish the scan pattern is 3.85 s?

The timing data given in Table 1 is extracted from original Windcube output files. The time in excess of 0.6667s is probably required to switch from one beam direction to the other. We do not know why the switching from one cardinal direction to the next occurs faster (0.72s) than switching to the vertical beam (0.97s). We added: *"The reason for the different times required to change the beam direction is not known to the authors."*

3. How do the authors get Equation 17?

We added a step in Eqs. 16 and 17 and described in a different way where these equations come from.

4. Are all the conditions provided in Page-9, L-10 and L-11 correct?

We checked these conditions carefully and believe that they are correct.

5. Is $k_{scan}$ (Equation-23) related to the scan pattern time? Or is it particularly related to the vertical beam (5th beam)? Due to the vertical beam, there are no measurements for horizontal wind speed. Could the authors make a comment on it?

$k_{scan}$ is not only relevant for the vertical beam but for all five beams because each line-of-sight velocity is updated after 3.85s. The lidar is therefore blind for all wind velocity components at $k_{scan}$. We added this aspect to the text and wrote *"That means turbulent fluctuations which occur with the same frequency cannot be detected by any of the Windcube's lidar beams."*

6. Page -11,L-19: what do the authors mean by, "for the combination of LOS2 and LOS4 it is similar"?

We mean that all what was described for the LOS1-LOS3 beam combination is true also for the LOS2-LOS4 beam combination but since this was unclear to the referee we deleted the quoted sentence.

7. Do the authors use Equation 25 (separation distance, $r_{rep}$) to calculate the time delay? How Equation 25 is different from Equation 24? It is not clear here.

The time delay is based on $r_{real}$ We fixed this mistake in the manuscript. We will use the new Figure 2 also to visualize the difference between $r_{real}$ (Eq. 24) and $r_{real,SQZ}$ (Eq. 25).

8. Second paragraph of Page 12, L-8 to L-12: "The methods of squeezing.." is irrelevant here. Remove the whole paragraph.

We followed the referee's recommendation to remove the whole paragraph.

9. Page-14, L-6: In case of three?

We change the formulation to "…and in the third case…"

10. Page-14, L-20 to L-25: Rewrite the sentences so that reader can understand the process: "In the process of reconstructing..."

We rewrote these sentences and also here we use the new Figure 2 to visualize the process of rearranging the measurements.

11. Page-17,L-1:The authors mention to explain the results: "the reason is that two different longitudinal separation distances are involved in the wind vector reconstruction process.... ".

However, in Table-1, authors showed that the second longitudinal separation distance is zero for the aligned flow. Could the authors explain what are they trying to explain here?

Table 2 is gives the **representative** longitudinal separation distances ($r_{rep}$) which are not the real longitudinal separation distances ($r_{real}$) as we explain at the end of section **2.4**

To improve the comprehensibility, we extended the caption of table 2, fixed the reference to section 2.4 and included a note that $r_{rep}$ is not equal $r_{real}$.

12.This article is particularly trying to solve an active industry problem and the article needs to be coherent in terms of explanation which I don't see in Section4. Explanation of the results is so vague that I get always lost.

Section 4 gives a description and interpretation of many different spectra. We intend to explain them as precise as possible. However, the high number of effects considered and the use of specific terms can create confusion.

We thank Prof. Emeis for his very encouraging and friendly review.

Stefan Emeis (Referee)

This manuscript gives an important analysis on how reliable are turbulent spectra derived from wind lidar measurements. This helps with the derivation of the necessary information for planning and operating wind turbines.

Main idea of this manuscript is sampling turbulence data from a computer-generated turbulence box. This gives the perfect opportunity to vary all relevant parameters necessary for an reliable assessment.

All this is presented in a very clear and concise style. Thus, I recommend to publish

We thank Anonymous Referee #3 very much for their comprehensive review of our manuscript. The feedback was of great help and we believe that the text is of a considerably higher quality in its revised version. We copied the referee comments and add our response, marked in red, after each point raised.

Anonymous Referee #3

Kelberlau and Mann present work towards an improved measurement of turbulence spectra from Doppler lidar DBS scans. They introduce a methodology to simulate the lidar measurements in a turbulence box which helps them to analyze the quality of the lidar measurements. With the method of squeezing that has been introduced in a previous study they achieve remarkable improvements by eliminating cross-contamination effects in the lidar measurements. They show that these improvements can only be achieved if the wind speed is aligned with the DBS scan and conclude that in all other conditions, the spectra cannot be corrected. I think this study provides very interesting analysis and important insights into DBS scanning. However, I found the manuscript hard to read in some parts, mostly because of unprecise language and variable definition. Despite this there are some other major concerns which I summarize in the general comments. I recommend the manuscript to be considered for publication in Wind Energy Science after major revisions.

0.1 General comments
• I think the introduction can be improved to better motivate the use of DBS scans for turbulence retrieval. There are many studies that use VAD-scans for this purpose. What is the advantage of using DBS? Please relate this to the work of Eberhard, Frehlich, Smalikho, Krishnamurthy, Bodini etc.
We extended the introduction by adding a definition of VAD and DBS, referring to previous research based on either of the two scanning strategies, and described the advantages when using DBS.
• Since this is a manuscript for Wind Energy Science, I think the authors should describe a little bit more how turbulence spectra can be used in practice for wind energy purposes. I think many wind energy experts are not very familiar with this topic. How exactly do they relate to IEC 61400-1
In the revised version of our manuscript we describe more accurately how turbulence spectra can be used to derive turbulence parameters for determining aerodynamic loads on wind turbines in accordance with IEC 61400-1.
• Please be very clear with directions, angle offsets and definitions. It is quite hard to follow the different coordinate systems that are used throughout the manuscript. A nomenclature of variables in the appendix would also help to serve this purpose.
We agree with the referee and added a nomenclature in the appendix. We also removed several inconsistencies in the variable names.
• Section 4 with the results stops at describing the differences between measurements and simulation in a qualitative way. I want to encourage the authors to consider adding a quantification of the error between lidar and sonic estimated turbulence parameters at least for the cases with aligned wind flow with the DBS scan. Also, many unknown behaviours are described without giving ideas about how to investigate this behaviour any further. This could be added to the conclusion.
A quantification of the systematic error of a DBS scanning wind lidar for different wind conditions is given in Sathe and Mann (2011). Determining the error based on our spectra would not create new knowledge. We therefore focus on understanding the different effects that influence the shape of the turbulence spectra. We extended section 5 (Conclusion) accordingly.
• The conclusion and outlook section is very short with a rather pessimistic ending stating that in most cases the turbulence spectra "should not be trusted". I think these findings should be related to the goal of wind site assessment and load prediction that is mentioned in the beginning. What are the prospects? How can this work help in future? What are alternative measurements that could be done for this purpose and what are the advantages/disadvantages compared to the method presented

in this study. One question that came to my mind is if a DBS strategy which adapts the beam direction to the wind direction could be used to overcome the problem of cross-contamination.

We extended the conclusion by adding the note "In no case should turbulence velocity spectra from DBS wind lidar be fitted to a turbulence model." to relate our findings to the goal of wind site assessment. An auto alignment of the beam orientation with wind direction is an interesting idea. We think such a technology would require knowledge about the wind direction at different height levels before the wind measurements are taken. Also, such a scanning strategy would not improve the ability to measure the $v$-component of the wind. We instead refer to the Windscanner multi-lidar technology and add an idea for a different modification of DBS scanning wind lidar, that includes deflecting the beams of one single lidar device, so that they intersect in a common area.

• I recommend some language copy-editing if the manuscript is accepted for publication.

0.2 Specific comments

• p.1, l.2f: The authors write that DBS lidars generate spectra. This is confusing, because it suggests that there is only one kind of velocity spectra and it is automatically produced by the lidar. I think the authors should be very clear from the beginning how these spectra are produced (i.e. from radial velocities, vertical stare or the retrieved wind vector).

Yes, we should be clear and changed the formulation to make clear from the beginning that we work with "*Spectra generated from reconstructed wind vectors of Doppler beam swinging (DBS) wind lidars*"

• p.1, l.7: The method of squeezing should maybe be briefly introduced, because it is not a well-known term in the community.

We add that the method of squeezing "*reduces the longitudinal separation distances between the measurement locations of the different lidar beams by introducing a time lag into the data processing*" because it is not a well-known term. A more detailed description of the method follows later in the manuscript.

• p.2, l.20ff: There exist some works that simulate lidar scans in LES fields (e.g. Stawiarski et al., 2015). What are differences / advantages of the method using the turbulence box. This could be described in more detail in Section 3.2.

LES simulations are more computationally expensive, especially since very long time series are required for deriving smooth spectra and small scale turbulent structures are not well represented in LES data (5-8 times the grid length). We added:

*"Sampling in a turbulence box is a method to simulate wind lidar measurements in very large computer-generated wind fields. The creation of such wind fields according to Mann (1998) requires less computational power than for example large eddy simulations (LES). LES was successfully used before to analyse coherent structures in wind fields (e.g. Stawiarsky et al., 2015) and wind profiles (e.g. Gasch et al., 2019) but predicting lidar derived turbulence velocity spectra requires much more turbulence data. An advantage of using LES is that Taylor's frozen turbulence hypothesis does not need to be applied but a drawback is that fine scale turbulence would be suppressed."*

• p.3, l.10: How is the time scale defined that divides the mean part from the turbulence part in the Reynolds decomposition?

We added the information that the time scale for averaging is ten minutes.

• p.7, ll. 9ff: I cannot follow how Eq. 16 and 17 are concluded from Eq. 13, 8 and 9. Also, it is defined in Sect. 2.1 that $u$ is the longitudinal wind component and $v$ the transversal, but now it seems that these are the meteorological conventions!?

In accordance with the comment of Anonymous Referee #1, we added a step in Eqs. 16 and 17 and described in a different way where these equations come from. $u$ and $v$ are always the longitudinal and transversal wind components. The introduction of $\alpha$ (see next point) might help seeing the relation between $u,v$ and $x,y$. more easily.

• p.8, Eqs.18-22: I think these equations could be presented in a more concise way for better readability. For example, $\Theta - \theta_0$ could easily be replaced by a single variable name and $\sigma_{2u}$ in Eq. 21

could be presented as a function of $u_{DBS}$. By the way, DBS as the variable subscript is a bit unfortunate. More than one letter in the subscript should not be italic.

We introduced the relative inflow angle $\alpha = \overline{\theta} - \theta_0$ and use it in all equations. We also rearranged many terms of Eqs. 13-22 for better readability. Eq. 21 does now include a representation of $u_{DBS}$. We changed all subscripts that are not a variable (DBS, SQZ, hor, long, lat, rep, real, res, scan) to appear in roman font.

• p.12, l.8: ZX300 was only briefly mentioned in the introduction. Maybe repeat here what is meant with the abbreviation.

We added that the ZX300 "*is a continuous-wave VAD scanning profiling lidar.*"

• p.12, l.12: What are .rtd-files. The file ending is not really important for the reader, but what kind of information they contain!

Mentioning the file ending makes it easier for readers that are familiar with the Windcube to know which files we mean because the different output files have different endings. For all other readers, we added that these files "*are standard output data files that contain the line-of-sight velocities of every single beam including their timing and carrier-to-noise ratio.*"

• p.12, l.25: The parameters should be introduced with their meaning.

That is true. We introduced the three model parameters "*the turbulence length scale L, the degree of anisotropy Γ, and the dissipation factor $\alpha \varepsilon^{\frac{2}{3}}$*".

• p.14, l.22: "project all focus points onto a vector..." I think this is unclear. What are the focus points in a pulsed lidar?

What we falsely named focus points are the centre points of the range gates along the lidar beams. We changed the expression to "*measurement locations*" in order to use an easy to read expression. In accordance with a comment of Anonymous Referee #1 we added a Figure 2 and extended the description of how the line-of-sight measurements are to be processed.

• p.14, l.27: Is a nearest neighbour method really the best solution? Would interpolation not be better (even though it would definitely also not be perfect in a turbulent flow)?

Both methods are not perfect. The nearest neighbour method has the advantage that the actual velocity values and as a consequence the total velocity variances remain unchanged. Interpolation would flatten the velocity peaks and reduce the variances slightly. We added that "*we reach that all measurement data is used with no change in velocity variance which would occur if interpolation would be applied*" to the description of our motivation to use the nearest neighbour method.

• p.14, ll.31ff: I recommend putting this description in a mathematical formula.

We followed the recommendation of the referee and put the description in a mathematical formula.

• p.15, l.6f: That the area under the power spectral density must equal the variance of the time signal follows from the Parseval's theorem and should always be checked and valid if power spectra are calculated. However, with the scaling with the wave number as it is done in Fig. 4, this does not apply. Please check and add the relevant literature and formula, if you mention it.

We believe that the statement is correct for our presentation of the spectra. We added Stull (1988) as a reference who writes "Semi-log presentation. By plotting f*S(f) vs. log f, the low frequency portions of the spectra are expanded along the abscissa. Also, the ordinate for the high frequency portions are enhanced because the spectral density is multiplied by frequency (see Fig 8.9d). Another excellent quality is that the area under any portion of the curve continues to be proportional to the variance."

Stull, R. B. (1988). *Some Mathematical & Conceptual Tools: Part 2. Time Series. An Introduction to Boundary Layer Meteorology, 295–345.* doi:10.1007/978-94-009-3027-8_8

• p.15,l.17: What is the "long axis"? Please be more specific. Also, define what is the "target spectra". Do these spectra contain volume averaging of the lidar?

We improved the description of the "target spectra" and state that they "*originate from sampling single points along the u-direction of the turbulence box with a frequency of 4 Hz.*" in the revised manuscript.

• Fig.4 Fig.5: The line styles of u- and v-component are hard to distinguish. It would be good to show the $k_{-5/3}$-slope in the plots to get an idea of how well the spectra fit to the inertial subrange theory.

Adding the $k^{-5/3}$-slope would not give additional information because all target spectra are already guaranteed to follow the $k^{-5/3}$-slope in the inertial subrange. In our semi-logarithmic presentation, the slope is not a straight line but a curve and we are worried to overload the plots by adding it.

• p.28, l.14: I think the number of the IEC-standard should appear in the reference.

We agree and added the number of the IEC standard in the list of references.

[revised manuscript text omitted]

 and

$$r_{\underline{rep,v}\text{rep},v} = \frac{\left|\sin\left(2(\overline{\Theta} - \theta_0)\right)\right| D}{\left|\cos\left(\overline{\Theta} - \theta_0\right)\right| + \left|\sin\left(\overline{\Theta} - \theta_0\right)\right|} \quad \frac{|\sin\alpha| \times r_{\text{long},13} + |-\cos\alpha| \times r_{\text{long},24}}{|\cos\alpha| + |\sin\alpha|} = \frac{|\sin(2\alpha)| D}{|\cos\alpha| + |\sin\alpha|} \tag{17}$$

for the $v$ component. The resulting  representative longitudinal separation distance values for the Windcube for four measurement heights 40 m, 60 m, 80 m, and 100 m and for three relative wind inflow angles  $\alpha = 0°$, 22.5°, and 45° are given in table 2. From these distances, the wave numbers at which we expect resonance can easily be determined with  $k_{\text{res}} = n\pi/r_{\text{rep}}$ where $n$ is an odd integer.

 Lateral separation distances $r_{\text{lat},ij}$ could be estimated in a similar way. But compared to longitudinal separations, the situation is different for wind velocity fluctuations measured at two laterally separated points. The spatial structure of turbulence leads to the wind velocity fluctuations becoming less correlated as the distance between the two measurement points increases. The coherence of the fluctuations is also weaker for small eddies than for large turbulent structures. That means that a turbulent structure can only be detected at two laterally separated points if the length scale of the turbulent structure is large

compared to the separation distance.  Lateral separation leads to contamination that occurs gradually without resonance points at specific wave numbers.

If the mean wind is aligned with two opposing lines-of-sight, e.g., blows in the LOS1 – LOS3 direction, then the $u$-component of the wind vector is reconstructed from two points that are only separated longitudinally. That means each turbulent structure is measured twice: once, when it passes the LOS1 location, and then some time later at the LOS3 location. Assuming frozen turbulence, measurements from points that are separated longitudinally are fully correlated, and resonance occurs at specific wave numbers. The $v$-component, on the contrary, is in this case reconstructed from the laterally separated points of LOS2 and LOS4, and a reduced correlation is found depending on the size of the turbulent structure and the separation distance. No specific resonance wave numbers are found. For a comprehensive description of the cross-contamination effects due to isolated longitudinal and isolated lateral separation, see Kelberlau and Mann (2019). Here we look at the more complex case when the mean wind inflow is not aligned with two opposing line-of-sight directions. Estimates of one horizontal wind velocity component can then be contaminated by contributions from both other wind velocity components. For a manual estimation of the cross-contamination effect for non-aligned inflow we first derive the lidar estimated wind vector component $u_{DBS}$ as a function of the real wind vector at all four measurement locations. When, Eqs. 8 and 9 are set into Eq. 13 we get

$$u_{DBS} = \frac{\cos(\overline{\Theta}-\theta_0)(\tilde{v}_{r_1}-\tilde{v}_{r_3})}{2\sin\phi}\frac{(\tilde{v}_{r_1}-\tilde{v}_{r_3})\cos\alpha}{2\sin\phi} + \frac{\sin(\overline{\Theta}-\theta_0)(\tilde{v}_{r_2}-\tilde{v}_{r_4})}{2\sin\phi}\frac{(\tilde{v}_{r_2}-\tilde{v}_{r_4})\sin\alpha}{2\sin\phi}. \tag{18}$$

We assume no line-of-sight averaging, so that $v_{r_i} = \tilde{v}_{r_i}$ and use Eqs. 4 and 5. After rearranging we get

$$u_{DBS} = \frac{\cos(\overline{\Theta}-\theta_0)}{2}\frac{\cos\alpha}{2}(-x_1+\cot(\phi)z_1\cot\phi-x_3-\cot(\phi)z-z_3\cot\phi)+\frac{\sin(\overline{\Theta}-\theta_0)}{2}\frac{\sin\alpha}{2}(-y_2+\cot(\phi)z_2\cot\phi-y_4-\cot(\phi)z- \tag{19}$$

After transferring the wind velocity components $x,y,z$ into the $u,v,w$ coordinate system we get

$$u_{DBS} = \frac{\cos(\overline{\Theta}-\theta_0)}{2}\frac{\cos\alpha}{2}(-\cos(\overline{\Theta}-\theta_0)u-u_1-\sin(\overline{\Theta}-\theta_0)v\cos\alpha-v_1-\cot(\phi)w\sin\alpha-w_1-\cos(\overline{\Theta}-\theta_0)u\cot\phi-u_3-\sin(\overline{\Theta}$$

$$+\frac{\sin(\overline{\Theta}-\theta_0)}{2}\frac{\sin\alpha}{2}(-\sin(\overline{\Theta}-\theta_0)u-u_2\sin\alpha+\cos(\overline{\Theta}-\theta_0)v_2-\cot(\phi)w\cos\alpha-w_2-\sin(\overline{\Theta}-\theta_0)u\cot\phi-u_4\sin\alpha- \tag{20}$$

With Eq. 3 we can describe the total lidar variance as a function of the wind vector fluctuations at the four measurement points as

$$\sigma_{u,DBS}^2 = \langle u'^2_{DBS}\rangle = \frac{1}{4}\left\langle\left(\cos(\overline{\Theta}-\theta_0)\left(\cos(\overline{\Theta}-\theta_0)u'_1\cos\alpha+\sin(\overline{\Theta}-\theta_0)v'_1\sin\alpha+\cot(\phi)w'_1\cot\phi+\cos(\overline{\Theta}-\theta_0)u'_3\cos\alpha+\sin(\right.\right.$$

$$+\sin(\overline{\Theta}-\theta_0)\left(\sin(\overline{\Theta}-\theta_0)u'_2-\cos(\overline{\Theta}-\theta_0)v'\sin\alpha-v'_2\cos\alpha+\cot(\phi)w'_2\cot\phi+\sin(\overline{\Theta}-\theta_0)u'_4-\cos(\overline{\Theta}-\theta_0)v\sin\alpha \tag{21}$$

A similar formula can be found for the transversal component

$$\sigma_{v,\mathrm{DBS}}{}^2 = \langle v'_{\mathrm{DBS}}{}^2 \rangle \approx \frac{1}{4}\Big\langle \Big(\sin(\overline{\Theta}-\theta_0)\big(\cos(\overline{\Theta}-\theta_0)u'_1\cos\alpha + \sin(\overline{\Theta}-\theta_0)v'_1\sin\alpha + \cot(\phi)w'_1\cot\phi + \cos(\overline{\Theta}-\theta_0)u'_3\cos\alpha + \sin(\overline{\Theta}$$

$$-\cos(\overline{\Theta}-\theta_0)\big(\sin(\overline{\Theta}-\theta_0)u'_2 - \cos(\overline{\Theta}-\theta_0)v\sin\alpha - v'_2\cos\alpha + \cot(\phi)w'_2\cot\phi + \sin(\overline{\Theta}-\theta_0)u'_4 - \cos(\overline{\Theta}-\theta_0)v\sin\alpha$$

$$(22)$$

Power spectral densities $F_{\mathrm{DBS}}$ at particular wavenumbers are composed of the same linear combinations of wind components as the total variances in Eqs. 21 and 22. These equations are thus helpful when analyzing the extent of cross contamination at particular wave numbers. As an example, we now take the case when the mean wind direction and one of the lines-of-sight create an angle of $45°$. We assume $\overline{\Theta} = 90°$ and $\theta_0 = 45°$ because this situation is found in the measurements described later in this study. However, the results are identical for all setups in which the relative wind inflow $\alpha = 45°$. In this case, LOS4 and LOS3 are separated purely longitudinally from LOS1 and LOS2, and LOS2 and LOS3 are separated purely laterally from LOS1 and LOS4 as shown in Figure 3. This opens up the possibility of determining the cross-contamination effect for four extreme conditions. These four extreme conditions are characterized by either full or no longitudinal resonance as well as either perfect or no lateral correlation. In the first case a) when no resonance occurs and the lateral correlation is perfect, we assume identical wind vectors at all four points. It accounts: $\boldsymbol{u'}_{1,a} = \boldsymbol{u'}_{2,a} = \boldsymbol{u'}_{3,a} = \boldsymbol{u'}_{4,a} = \boldsymbol{u'}_{\mathrm{I}}$. In the second case b) when no resonance occurs but the lateral correlation is zero, it accounts: $\boldsymbol{u'}_{1,b} = \boldsymbol{u'}_{4,b} = \boldsymbol{u'}_{\mathrm{I}}$ and $\boldsymbol{u'}_{2,b} = \boldsymbol{u'}_{4,b} = \boldsymbol{u'}_{\mathrm{II}}$ where $\boldsymbol{u'}_{\mathrm{I}}$ and $\boldsymbol{u'}_{\mathrm{II}}$ are independent vectors. In the third case c) resonance between the longitudinally separated points occurs and the fluctuations at laterally separated points are perfectly correlated. It accounts: $\boldsymbol{u'}_{1,c} = \boldsymbol{u'}_{2,c} = -\boldsymbol{u'}_{3,c} = -\boldsymbol{u'}_{4,c} = \boldsymbol{u'}_{\mathrm{I}}$. The fourth case d) is characterized by longitudinal resonance and zero lateral correlation. It accounts: $\boldsymbol{u'}_{1,d} = -\boldsymbol{u'}_{4,d} = \boldsymbol{u'}_{\mathrm{I}}$ and $\boldsymbol{u'}_{2,d} = -\boldsymbol{u'}_{3,d} = \boldsymbol{u'}_{\mathrm{II}}$ where $\boldsymbol{u'}_{\mathrm{I}}$ and $\boldsymbol{u'}_{\mathrm{II}}$ are independent vectors. Figure 3 gives an overview of the conditions we assume for these four cases a) to d). With these assumptions, Eq. 21 provides the lidar estimates of the power spectral density values $F_{u,\mathrm{DBS}}$ as linear combinations of the spectral values of the three wind components $F_u$, $F_v$ and $F_w$, as shown in the lower half of table 3. The resulting linear combinations of power spectral densities that compose the lidar-measured $u$ and $v$-components of turbulence for the case with $\alpha = 0°$ are shown in the upper half of the same table.

Table 3 can be read as follows. First, choose the aligned ($\alpha = 0°$) or non-aligned case ($\alpha = 45°$). Then select the wind component of interest: $F_{u,\mathrm{DBS}}$ or $F_{v,\mathrm{DBS}}$. Next, decide if the situation with or without resonance is more relevant for the wave number of interest. Finally, select a block of values that either represents the case with perfect lateral correlation or that assumes laterally uncorrelated fluctuations. The sum of the variances of the wind components multiplied by the values given in this block is the theoretical lidar derived variance of the selected component. It is usually unclear to which degree the fluctuations are correlated but the table can still be used for rough estimations. If you look for example at the resonance case for $u$ you will find that the lidar does not detect longitudinal wind fluctuations at all, while the lidar estimated $u$-variance $F_{u,\mathrm{DBS}}$ is composed of a weakened $v$-signal of between 0.00 and 0.50 times the real $v$-fluctuations and an amplified $w$-signal of between 3.54 and 7.07 times the real $w$-fluctuations depending on the degree of lateral correlation. The values given in the table can explain many of the effects we later see in the lidar derived spectra for non-aligned inflow.

[Figure]

a) No resonance, laterally correlated
b) No resonance, laterally uncorrelated
c) Resonance, laterally correlated
d) Resonance, laterally uncorrelated

**Figure 3.** Overview of the assumptions made to determine the cross-contamination values listed in Table 3. In cases with no resonance, the wind vectors $u'_{\text{I,II}}$ are identical at the longitudinally separated measurement points. In resonance cases they have an opposite sign. In cases with laterally correlated velocities, the wind vectors at laterally separated measurement points are identical. And in cases with no correlation at points that are laterally separated the wind vectors $u'_{\text{I}}$ and $u'_{\text{II}}$ are independent.

Table 1 shows that the radial velocity for  each line-of-sight is determined not continuously but once every $3.85\,\text{s}$. That means  turbulent fluctuations which occur with a corresponding frequency cannot be detected by  any of the Windcube's lidar beams. The respective wave numbers are

$$k_{\underline{\text{scan}}} = \frac{2\pi}{U \cdot 3.85\,\text{s}}. \tag{23}$$

[revised manuscript text omitted]

---

## Author Response (AR2)

We thank Anonymous Referee #3 a lot for their efforts in reviewing also the revised version of our manuscript. We agree with all points raised and changed the text accordingly.

1. My only remaining concern is Eq. 25 and the statement about variance equalling the area under the curves in Figs.5ff. The authors respond to my comment with a quote from Stull 1988, which states that "the area under any portion of the curve continues to be proportional to the variance". I agree to the statement in Stull, but it does not say that the area under the curve equals the variance. It states that any portion is proportional. The integral under $F(k)$ is the variance (see Eq.8.6.2a in Stull 1988). It becomes very clear when you look at the units of the x- and y-axis in Fig.5. The unit of the y-axis ($k F(k)$) is m^2 s^{-2}, which is the unit of the variance. Integrating over the wave number yields the unit m s^{-2} which is not the unit of a variance.

   We agree with the reviewer's statement and their argumentation and changed our text to: "Displayed like this, any portion of the area under the spectra for a range of wave numbers is proportional to the variance of the signal in this wave number range \citep{stull.88.8}."

2. Additionally, I think that Eq.25 is a bit irritating, because it introduces a function of the wave number, but on the right hand side of the equation, the sampling frequency appears. I think it would be clearer to use $k$ in the right-hand side and maybe introduce $k$ as a function of $U$ and $f_s$. In p.16,l.19 I would recommend to write in the text that for cross spectra, the real part of $F_{ij}$ is used.

   We changed Eq.25 accordingly and introduce the sampling wave number $k_s$ as a function of $U$ and $f_s$ just after the equation and added it to the nomenclature. For the cross-spectra we wrote that "we use the real part of $F_{ij}$."

In addition to the reviewer's comments we changed all occurrences of "wavenumber" to "wave number" and all occurrences of "cross spectrum" to "cross-spectrum". All changes are highlighted in the attached latexdiff version of the manuscript.

[revised manuscript text omitted]